

# The giant graviton expansion in $AdS_5 \times S^5$

Giorgos Eleftheriou[1⋆], Sameer Murthy[1,2†] and Martí Rosselló[3‡]

**1** Department of Mathematics, King's College London, The Strand, London WC2R 2LS, UK
**2** School of Natural Sciences, Institute for Advanced Study, Princeton, NJ 08540, USA
**3** Institute of Mathematics, Academia Sinica, Taipei, Taiwan

⋆ geleftheriou4@gmail.com , † sameer.murthy@kcl.ac.uk ,
‡ marti@gate.sinica.edu.tw

## Abstract

The superconformal index of $\frac{1}{2}$-BPS states of $\mathcal{N} = 4$ $U(N)$ super Yang-Mills theory has a known infinite $q$-series expression with successive terms suppressed by $q^N$. We derive a holographic bulk interpretation of this series by evaluating the corresponding functional integral in the dual $AdS_5 \times S^5$. The integral localizes to a product of small fluctuations of the vacuum and of the collective modes of an arbitrary number of giant-gravitons wrapping an $S^3$ of maximal size inside the $S^5$. The quantum mechanics of the small fluctuations of one maximal giant is described by a supersymmetric version of the Landau problem. The quadratic fluctuation determinant reduces to a sum over the supersymmetric ground states, and precisely reproduces the first non-trivial term in the infinite series. Further, we show that the terms corresponding to multiple giants are obtained precisely by the matrix versions of the above super-quantum-mechanics.

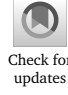

## 1   Introduction and summary

**Aim of the paper**

We consider four-dimensional $\mathcal{N} = 4$ $U(N)$ super Yang-Mills (SYM) theory on $S^3$, and study the supersymmetric index that enumerates $\frac{1}{2}$-BPS states (or, equivalently by the state-operator correspondence, local operators) in this theory. The index is defined as the trace

$$I_N(q) = \text{Tr}_{\mathcal{H}^N_{\frac{1}{2}\text{-BPS}}} (-1)^F q^R, \qquad (1)$$

where $\mathcal{H}^N_{\frac{1}{2}\text{-BPS}}$ is the $\frac{1}{2}$-BPS Hilbert space consisting of states that are annihilated by a certain 16 of the 32 supercharges of the theory, $(-1)^F$ is the fermion number operator, and $R$ is a Cartan generator of the $su(4)$ R-symmetry algebra of the theory. The $\frac{1}{2}$-BPS operators in the free theory are arbitrary holomorphic functions of one of the matrix-valued complex scalars of the SYM theory with $R = 1$. The ring of such functions is generated freely by the first $N$ powers of the matrix, and the index is therefore given by

$$I_N(q) = \frac{1}{(q)_N}, \qquad (2)$$

in terms of the $q$-Pochhammer symbol $(q)_n := \prod_{j=1}^{n}(1 - q^j)$. A well-known mathematical result (see Appendix A for a derivation) is that this index can be rewritten as follows

$$I_N(q) = I_\infty(q) \sum_{m=0}^{\infty} (-1)^m \frac{q^{m(m+1)/2}}{(q)_m} q^{mN}. \qquad (3)$$

The aim of this paper is to provide a bulk interpretation of the right-hand side of this formula in terms of wrapped D-branes in the dual Type IIB theory on AdS$_5 \times S^5$, i.e. giant gravitons [1].

It should be noted that the interpretation of giant gravitons in the AdS/CFT correspondence has been discussed from different points of view in the past [2–17] and, in particular, canonical quantization of the moduli space of bulk D-branes [18], using a certain symplectic form, directly leads to the product formula on the right-hand side of (2) [19,20]. The present work, instead, identifies each term of the right-hand side of (3) from the bulk point of view, in the spirit of the recently-discussed "giant graviton expansion" [21–23].

**Broader context**

More generally, one can consider the superconformal index in a $d$-dimensional superconformal field theory (SCFT$_d$) on a compact spatial manifold, which is defined as a generalized Witten index, as in (1), of BPS states that are annihilated by some number of supercharges of the theory. In its simplest form it depends on one parameter $q$ which is the fugacity for a charge $R$ that commutes with the preserved supercharges, and can be written as a $q$-expansion,

$$\text{Tr}_{\mathcal{H}_{\text{BPS}}} (-1)^F q^R = \sum_{r} d(r) q^r. \qquad (4)$$

The coefficients $d(r)$ have the interpretation of the (indexed) number of states with charge $R = r$. One can also have refinements of (4) with other fugacities coupling to other commuting charges.

The index (4) is invariant under small changes of couplings of the theory [24]. Modulo issues of wall-crossing (which will not appear in this paper) the value of the index at zero coupling equals that at strong coupling, where it should have a gravitational interpretation in the dual AdS space according to the AdS/CFT correspondence. At zero coupling, the index of states preserving various fractions of supercharges can be calculated using either an explicit enumeration of operators, or by localizing the corresponding path integral to a $U(N)$ matrix model [25, 26], and these serve as exact predictions for gravitational calculations.

Taking the $\mathcal{N} = 4$ $U(N)$ super Yang-Mills (SYM) theory for concreteness, the scale of gravitational phenomena in the dual AdS$_5$ space is set by the gravitational coupling constant $G = 1/N^2$. It has been known since the early days of the AdS/CFT correspondence that for very small charges $r \ll N$, the numbers $d(r)$ can be interpreted as the indexed number of supergravitons or perturbative closed strings i.e. single-trace operators in the gauge theory.[1] In recent years it has been shown that for very large charges $r = O(N^2)$, $\log |d(r)|$ calculates the entropy of supersymmetric black holes in the dual AdS$_5$ [27–32]. In the present context we are concerned with charges of intermediate scales $r = O(N)$, which is the characteristic scale of wrapped D-branes in AdS$_5$ space.

**The giant graviton expansion**

A detailed interpretation of formulas of the type (3) in terms of wrapped branes, for different indices in string and M-theory, was advocated in a series of works by Imamura and collaborators [21, 33–35], with numerical evidence for small numbers of branes.[2] The idea is that the $m^{\text{th}}$ term in the formula (3) is associated with $m$ wrapped D-branes in the bulk of AdS space. The term $q^{mN}$ is supposed to be the ground state energy of these D-branes which is consistent with the fact that the tension of D-branes scales as $1/N$ for large $N$, and that the bulk interpretation of wrapped D-branes are determinant operators in the gauge theory [36]. The rest of the $m^{\text{th}}$ term is interpreted as arising from fluctuations of $m$ branes. This idea was taken forward in the papers [22, 37] which presented further numerical evidence, and discussed the determinantal nature of gauge-theory operators which contribute to such an expansion, and gave the current name to the expansion. This approach was further studied by residue methods in [38]. In all these results, one typically needs at least two fugacities, one of which dominates the expansion and the second is a perturbative parameter.

In [23] it was shown that an expansion of the giant graviton form arises for any unitary matrix model that arises in counting invariants of unitary groups (which includes superconformal indices in gauge theory with one or many fugacities). More precisely, one has, for the $U(N)$ integral $Z_N(q)$ depending even on a single fugacity,

$$\frac{Z_N(q)}{Z_\infty(q)} = \sum_{m=0}^{\infty} G_N^{(m)}(q), \qquad G_N^{(m)}(q) \in q^{\alpha mN + \beta} \mathbb{Z}[[q]], \tag{5}$$

with $\alpha$ a constant and $\beta$ a polynomial in $m$. This formula is derived by introducing an auxiliary problem which admits a free-fermion representation [39], and then writing the matrix integral as average over the couplings of the free fermion theory. The free fermionic nature of the problem gives a natural determinantal expansion which explains the form of the expansion and

---

[1]In fact this interpretation is true for all indices all the way until $r = \alpha N$, where $\alpha$ is a fixed O(1) non-zero number depending on the details of the index [23].

[2]For the $\frac{1}{8}$-BPS Schur index, a formula of the form of (3) was derived in [16, 17] and it was suggested there that it could be interpreted in terms of giant gravitons in the bulk.

gives a formula for each term $G_N^{(m)}(q)$. The first and second terms were explicitly calculated in [23] and [40], respectively. An interpretation of this free fermion expansion in terms of an instanton expansion was discussed in [41].

It is important to note that an expansion of the type (5) is not unique, since say the first and second giant terms can overlap after the first $\alpha N$ terms. In fact the concrete formulas in [21], [22], and [23] all differ at this level.[3] This should be interpreted as the ambiguity in a choice of basis of operators in the gauge theory ("operator mixing issue"). It is therefore important to have a good bulk interpretation of these formulas so as to have a starting point for the basis of operators. For this we return to the simplest case of the $\frac{1}{2}$-BPS index.

**Approach and results of this paper**

Our approach to obtain the bulk interpretation of (3) is to directly quantize the space of $\frac{1}{2}$-BPS giants in the bulk as a Euclidean functional integral over the configuration space of giants in a "first-quantized" treatment. In other words, we want to integrate over the moduli space of an arbitrary number of $\frac{1}{2}$-BPS giants and include the fluctuations of the giants at every point in moduli space. The time direction runs in a circle in the Euclidean formalism and fermions have periodic boundary conditions in accord with the supersymmetric index. The set-up is similar to a gas of instantons [43,44], except that there is no sense in which the giants are dilute.[4] (In fact, as we see below, the main contributions come from groups of giants sitting on top of one another at a single point.) What allows us to perform this calculation is the use of supersymmetric localization, which localizes the integral to the fixed points of a certain complex supercharge $Q$.

There are two types of $\frac{1}{2}$-BPS wrapped branes in $AdS_5 \times S^5$. The first type are D3-branes that sit at the center of $AdS_5$, wrap an $S^3$ inside the $S^5$ and rotate around a circle of $S^5$ at the speed of light. The semi-classical moduli space of these configurations is parameterized by the size of the giants. This size takes values from zero, when the giants are point-like and identified with ordinary gravitons, to a maximum value, when the giants wrap a maximal $S^3$ inside the $S^5$. The second type, called "dual giants" wrap an $S^3$ inside $AdS_5$, and rotate along an equatorial circle of $S^5$ [2,4]. There is no classical bound on the size of the dual giants, but since each brane carries one unit of charge, it is expected that one cannot place more than $N$ dual giants [12,46].

In order to localize the functional integral, we choose a supercharge which obeys the algebra $\{Q, \overline{Q}\} = H - R$, where $H$ is the time-translations generator and $R$ is the generator of translations in the angular coordinate $\phi$ parameterizing the circle on $S^5$ along which the giants move. As it turns out, the fixed points of $Q$ in the Euclidean theory are precisely the maximal giants, whose number $m = 0, 1, 2, \ldots$ can be arbitrary. Summing over this number explains the sum over $m$ in the formula (3).

Around each fixed point, we then perform the supersymmetric functional integral of quadratic fluctuations of the maximal giants. The fluctuations around any number of giants are of two types, as in an instanton gas: the fluctuations of the background field theory—in our case supergravity on $AdS_5 \times S^5$, and the fluctuations of the collective modes of the instantons—in our case $m$ giants. The supersymmetric integral over quadratic fluctuations factorizes into the index of these pieces. The index over the supergravity modes is precisely equal to $I_\infty$ as can be shown from multiple points of view [23,26,47], so we are left to explain the ratio $I_N/I_\infty$ in (3) from the fluctuations of the giant gravitons in $AdS_5 \times S^5$.

---

[3]A prescription to relate the different series term-by-term has been discussed in [42].

[4]A similar approach to calculations on M-branes has been taken recently in [45].

The quadratic fluctuations around a single giant at a generic point in moduli space were analyzed in [3] and leads to a gapped spectrum of excitations. However, that analysis breaks down precisely for the maximal giants in which case, as we see below, one has massless excitations. As it turns out, these excitations are governed by a supersymmetric version of a particle moving in two dimensions in a constant magnetic field, i.e. the Landau problem. In addition one has a delta-function valued flux line at the origin coming from the background giant, which can also be treated formally as a gauge transformation of the Landau problem, or as the Aharonov-Bohm effect.

The one-loop fluctuation determinant around the $m = 1$ fixed-point therefore reduces to the evaluation of an index of the type (1) on the Hilbert space of states of the supersymmetric particle in a magnetic field. In particular, we do not need the (unknown) non-linear supersymmetric Lagrangian of multiple branes, but only that of quadratic fluctuations. Upon summing over the infinite set of supersymmetric ground states, and taking into account the fermionic zero modes, one obtains precisely the infinite sum $-\sum_{j=1}^{\infty} q^j$, which is precisely the expected result for one giant from (3). The analogous calculation for $m$ giants is given by a matrix version of the same problem as in [8, 48], and gives precisely the result expected from the $m^{\text{th}}$ term in (3).

**Comments**

1. It is important for our derivation that we are calculating a supersymmetric index. Firstly, this is the reason it is protected and can be interpreted in exact terms in the bulk. Secondly, it allows us to use the technique of supersymmetric localization to calculate the functional integral. Thirdly, although the coefficients $d(r)$ for the $\frac{1}{2}$-BPS index (2) are all positive, the giant-graviton formula (3) has negative signs which are explained by the fermionic nature of the supersymmetric states contributing to the fluctuation determinant.

2. The boundary calculation as well as our giant-graviton calculation of the index involves fluctuations of D-branes in flat space and in AdS space, respectively. Relatedly, as was explained in [23], the giant graviton expansion should be considered as the second open string theory in the open-closed-open duality as in [49, 50]. It is, however, important to note that our derivation of the giant graviton expansion from the bulk is a Euclidean functional integral calculation. The $m^{\text{th}}$ term is interpreted as the contribution of the saddle with $m$ giants, which are Euclidean instantons rather than sectors of Hilbert space of SYM.[5] The fluctuation determinant around the saddle allows for a factorization and for fermionic contributions. In contrast, in the Hilbert space interpretation of $\frac{1}{2}$-BPS states in CFT$_4$/AdS$_5$, the D-branes are interpreted as droplets of the Fermi liquid (derived from the single matrix model in the boundary [8] and the LLM geometries [51] in the bulk) and always have a positive degeneracy.[6]

3. The $\frac{1}{2}$-BPS index is clearly the simplest of many supersymmetric indices arising in different manifestations of the AdS/CFT duality. The ideas of this paper should apply to these more general indices.[7] For a given observable, one can localize the corresponding functional integral using different choices of the supercharge or the localizing term. It is possible that different such choices lead to different formulas for the giant-graviton expansion that exist in the literature, as mentioned above.

---

[5]There is an auxiliary construction of the Hilbert space of the fluctuations around a saddle in our calculation, but this is not a priori related to the Hilbert space of SYM theory.

[6]We thank J. Maldacena for emphasizing this point to us.

[7]They could also be used to study related problems as in [52–54].

4. If we express the expansion in terms of $q^{-1}$ rather than $q$, the first giant term in the expansion (3) is $1/(1-q^{-1})$, which looks like the bosonic partition sum of one oscillator with $R = -1$ (a similar statement is true for $m$ giants). Of course no such mode exists in the physical Hilbert space. Instead, this fact was interpreted in [21,22,35] as the analytic continuation from $|q^{-1}| < 1$ to $|q| < 1$ of a formal calculation involving D-branes. Here we explain the result directly in terms of the physics of D-branes and their fluctuations.

**Note added:**    While this paper was being prepared, we received [55] on the arXiv, which has some overlap with the current paper. The precise relation between our path integral approach and the Hilbert space interpretation of [55] in terms of open strings and ghosts is unclear to us.

**Plan of paper:**    In Section 2 we review the semiclassical description of the giant gravitons as D3-branes wrapping cycles. Further, we set up the theory of small fluctuations of the maximal giant in terms of a particle in a magnetic field. In Section 3 we set up and solve the localization problem for one giant, including the critical points and the one-loop determinants. In Section 4 we discuss multiple giants and the diagonalization of the holomorphic sector, and reach the expected giant graviton expansion from the bulk. In three appendices we review, respectively, the mathematical derivation of the $\frac{1}{2}$-BPS giant graviton expansion, the relation between the small fluctuations of the brane and the Landau problem, and the superalgebra on the brane.

## 2  The description of the bulk D-branes

In this section we collect some of the properties of the giant gravitons in $AdS_5 \times S^5$ that are $\frac{1}{2}$-BPS, i.e. preserve 16 of the 32 supercharges [2]. We focus on the physical properties of the bosonic coordinates in this section. In the following section we discuss aspects of supersymmetry that are important to our problem.

### 2.1  The background $AdS_5 \times S^5$

We follow the notation of [2]. The background spacetime is a direct product of $AdS_5$ and $S^5$

$$ds^2 = ds^2_{AdS_5} + ds^2_{S^5}, \tag{6}$$

with the respective line elements given by

$$ds^2_{AdS_5} = -\left(1 + \frac{r^2}{L^2}\right) dt^2 + \left(1 + \frac{r^2}{L^2}\right)^{-1} dr^2 + r^2 d\Omega_3^2, \tag{7}$$

and

$$ds^2_{S^5} = L^2\left(d\theta^2 + \cos^2\theta \, d\phi^2 + \sin^2\theta \, d\Omega_3^2\right). \tag{8}$$

Here $L$ is the radius of the $AdS_5$ as well as the $S^5$. The parameterization of $S^5$ is such that the slice for a fixed value of $\theta$ is of the form $S^1 \times S^3$. At $\theta = 0$ the $S^3$ reduces to a point and the $S^1$ reaches its maximum radius $L$. At $\theta = \pi/2$, the $S^3$ reaches its maximum radius $L$ and the $S^1$ reduces to a point. The value $\theta = \pi/2$ is thus a fixed point of the vector field $\partial_\phi$.

The five-form field strength is self-dual and is proportional to the volume form in $AdS_5$ as well as in $S^5$. We use the following angular coordinates for the $S^3$ inside $S^5$,

$$d\Omega_3^2 = d\chi_1^2 + \sin^2\chi_1\left(d\chi_2^2 + \sin^2\chi_2 \, d\chi_3^2\right). \tag{9}$$

The 4-form potential on the $S^5$ can be taken to be

$$A_{\phi \chi_1 \chi_2 \chi_3} \; = \; L^4 \sin^4 \theta \sin^2 \chi_1 \sin \chi_2 \; = \; L^4 \sin^4 \theta \sqrt{g_{S^3}} \,, \tag{10}$$

so that it yields a field strength proportional to the volume form $dA_{S^5} = F_{S^5} = \frac{4}{L} \mathrm{Vol}(S^5)$.

Now consider a $D3$-brane in $S^5$ with worldvolume coordinates $\sigma_i$, $i = 0, 1, 2, 3$. Its bosonic action is given by

$$S_3 \; = \; -T_3 \int d^4 \sigma \, \sqrt{-g} \; + \; T_3 \int P\left[A^{(4)}\right] . \tag{11}$$

Here $T_3$ is the tension of the D3-brane, $g$ is the pullback of the spacetime metric $G$ and $P\left[A^{(4)}\right]$ is the pullback of the 4-form (10). Writing the embedding as $X^M(\sigma^i)$ and the above background metric as $ds^2 = G_{MN} dX^M dX^N$, we have the expression

$$g_{ij} \; = \; \partial_i X^M \partial_j X^N G_{MN} \,, \tag{12}$$

and

$$\int P\left[A^{(4)}\right] = \int d^4 \sigma \, \frac{1}{4!} \, \varepsilon^{i_0 i_1 i_2 i_3} \, \partial_{i_0} X^{M_0} \dots \partial_{i_3} X^{M_3} A_{M_0 M_1 M_2 M_3} \,, \tag{13}$$

with $\varepsilon^{0123} = 1$.

## 2.2 Semiclassical giant gravitons

Now we discuss the D3-branes that wrap an $S^3 \subset S^5$. They have a spherical symmetry corresponding to rotations of the worldvolume $S^3$, and another spherical symmetry corresponding to rotations of the $S^3 \subset S^5$. We can describe them by choosing a static-like gauge for the D3-brane embedding,

$$\sigma_0 \; = \; t \,, \qquad \sigma_i \; = \; \chi_i \,, \quad i = 1, 2, 3 \,, \tag{14}$$

and make the following Ansatz,

$$\phi \; = \; \phi(t) \,, \qquad \theta \; = \; \theta(t) \,, \qquad r \; = \; r(t) \,, \tag{15}$$

with the rest of the coordinates being constant. This gives, with $\dot{} = \frac{d}{dt}$

$$g_{tt} \; = \; -\left(1 + \frac{r^2}{L^2}\right) + L^2 \left(\cos^2 \theta \, \dot{\phi}^2 + \dot{\theta}^2 + \frac{\dot{r}^2}{L^2} \left(1 + \frac{r^2}{L^2}\right)^{-1}\right) , \tag{16}$$

$$g_{ij} \; = \; L^2 \sin^2 \theta \, g_{S^3 ij} \,, \tag{17}$$

where $g_{S^3 ij}$ is the $S^3$ metric (9) in $\sigma_i$ coordinates.

Upon evaluating the action of the brane (11) for this configuration, and then integrating over the angular part, one obtains the following one-dimensional Lagrangian

$$\mathcal{L}_{\mathrm{giant}} \; = \; \frac{N}{L}\left(-\sin^3 \theta \sqrt{1 + \frac{r^2}{L^2} - L^2 \left(\cos^2 \theta \, \dot{\phi}^2 + \dot{\theta}^2 + \frac{\dot{r}^2}{L^2} \left(1 + \frac{r^2}{L^2}\right)^{-1}\right)} + L \sin^4 \theta \, \dot{\phi}\right), \tag{18}$$

where we have used the relation $A_3 T_3 = N/L^4$ between the flux and the tension of the brane with $A_3 = 2\pi^2$ the area of a unit 3-sphere.

Giant gravitons are solutions to the equations of motion of the above Lagrangian which move at the speed of light around a circle in the $S^5$. They are described by

$$r \; = \; 0 \,, \qquad \dot{\phi} \; = \; \frac{1}{L} \,, \qquad \theta \; = \; \theta_0 \,, \qquad \theta_0 \in [0, \pi/2] \,. \tag{19}$$

The constant value $\theta_0$ parameterizes the moduli space of giants. The value $\theta_0 = \pi/2$ corresponds to the brane with largest size (equal to $L$) and largest angular momentum (equal to $N$). This solution, called the maximal giant, does not have kinetic rotation and its angular momentum appears from the Chern-Simons coupling of the brane to the background RR-field [2].

## 2.3 Small fluctuations and the Landau problem

Now we consider the small fluctuations around the maximal giant, i.e. $\theta_0 = \pi/2$. The analysis of small fluctuations of the Lagrangian (18) for a generic value of $\theta_0$ was performed in [56], which we follow.

The results are as follows. Firstly, the fluctuations in all the coordinates except the fluctuations of $\theta, \phi$ are gapped. For example, the $r$-fluctuation is described by the oscillator

$$\frac{N}{L}\left(\frac{1}{2}\dot{r}^2 - \frac{1}{2}\frac{r^2}{L^2}\right).\tag{20}$$

Similarly, the angular fluctuations in the $S^3 \subset \mathrm{AdS}_5$ are described by massive spherical harmonics, and the internal gauge-field fluctuations of the brane world-volume in the $S^3 \subset S^5$ are also massive spherical vector harmonics. The scale of both these sets of fluctuations is set by the curvature of the $\mathrm{AdS}_5 \times S^5$. In the supersymmetric theory that we consider below, the fermionic partners of these bosonic fluctuations are also gapped. This part of the quantum-mechanical problem has a unique ground state.

Therefore we focus on the fluctuations of the center of mass on the $S^2$ with coordinates $(\theta, \phi)$. We expand around the solution (19) as

$$\phi(t) = \tfrac{1}{L}t + \epsilon\,\delta\phi(t), \qquad \theta(t) = \theta_0 + \epsilon\,\delta\theta(t).\tag{21}$$

The Lagrangian (18) to quadratic order in $\epsilon$ takes the form

$$\frac{L}{N}\mathcal{L}_{\mathrm{giant}} = \epsilon L \sin^2(\theta_0)\,\dot{\delta\phi}(t)$$
$$+ \epsilon^2 L^2\left(\frac{1}{2}\sin^2(\theta_0)\,\dot{\delta\theta}(t)^2 + \frac{2}{L}\cos(\theta_0)\,\delta\theta(t)\sin(\theta_0)\dot{\delta\phi}(t) + \frac{1}{2}\cos^2(\theta_0)\,\dot{\delta\phi}(t)^2\right) + \dots\tag{22}$$

We see that for a generic $\theta_0$ the fluctuations are also massive, as explained in [56]. Note, however, that when the giant graviton has maximal size, $\theta_0 = \pi/2$, the previous expansion of the Lagrangian becomes

$$\epsilon\, L\,\dot{\delta\phi}(t) + \frac{1}{2}\epsilon^2 L^2\,\dot{\delta\theta}(t)^2 + \dots,\tag{23}$$

and, in particular, there is no quadratic term in $\dot{\delta\phi}(t)$.[8] This is because one reaches the pole of the sphere at that point, and $\phi$ effectively becomes an angular direction, while fluctuations of $\theta$ around $\pi/2$ is a radial direction. Therefore we need to treat it as such and keep higher order terms that couple the two variables.

It is useful to define the coordinates

$$\rho = \frac{\pi}{2} - \theta, \qquad \dot{\varphi} = \frac{1}{L} - \dot{\phi},\tag{24}$$

which take the values $\rho = 0$, $\dot{\varphi} = 0$ for the maximal giant. As mentioned above, we need to keep terms of the type $\rho^2\dot{\varphi}^2$ in these polar coordinates, since they are part of the kinetic energy. The Lagrangian of small fluctuations in the $(\rho, \varphi)$ directions is

$$\mathcal{L}_{\mathrm{max}}^{(2)} = \frac{N}{L}\left(\frac{1}{2}L^2\rho^2\dot{\varphi}^2 + \frac{1}{2}L^2\dot{\rho}^2 + L\rho^2\dot{\varphi} - L\dot{\varphi}\right).[9]\tag{25}$$

---

[8]Indeed, the equations of motion presented in [56] become singular when expanded around $\theta_0 = \pi/2$.

[9]The same procedure can be performed in a unified way for giant gravitons in $\mathrm{AdS}_4 \times S^7$ and $\mathrm{AdS}_7 \times S^4$, which are M5 and M2 branes wrapping the spheres $S^5 \subset S^7$ and $S^2 \subset S^4$, respectively, and rotating around a transverse circle. These configurations were studied in e.g. [1,2,4,56], and the analysis is very similar to the $\mathrm{AdS}_5 \times S^5$ case.

This Lagrangian (25) describes a particle moving in a two-dimensional plane in a constant transverse magnetic field with an infinitely thin solenoid centered at the origin. In order to see this, it is useful to consider the change from polar to Cartesian coordinates,

$$L^2 \rho^2 = x^2 + y^2, \qquad \varphi = \arctan y/x. \tag{27}$$

The Lagrangian (25) for the quadratic fluctuations on the $S^2$ in these coordinates is

$$\mathcal{L}_{\mathrm{max}}^{(2)} = \frac{N}{L}\left(\frac{1}{2}\left(\dot{x}^2 + \dot{y}^2\right) + \frac{1}{L}(x\dot{y} - y\dot{x})\left(1 - \frac{L^2}{x^2 + y^2}\right)\right). \tag{28}$$

Note that the term $-N(x\dot{y} - y\dot{x})/(x^2 + y^2)$ comes from the linear term $-N\dot{\varphi}$ in (25). We can write this Lagrangian as

$$\mathcal{L}_{\mathrm{max}}^{(2)} = \frac{1}{2}\frac{N}{L}\left(\dot{x}^2 + \dot{y}^2\right) + \dot{\vec{x}} \cdot \vec{A}, \tag{29}$$

with

$$\vec{A} = \vec{A}_1 + \vec{A}_2, \qquad \vec{A}_1 = \frac{N}{L^2}\left(x\hat{y} - y\hat{x}\right), \qquad \vec{A}_2 = -\frac{N}{x^2 + y^2}\left(x\hat{y} - y\hat{x}\right). \tag{30}$$

Embedding the $x$-$y$ plane in three-dimensional flat space with coordinates $x, y, z$, the magnetic field is as follows. The first term in (30) yields

$$\vec{B}_1 = \frac{N}{L^2}\vec{\nabla} \times (x\hat{y} - y\hat{x}) = 2\frac{N}{L^2}\hat{z}. \tag{31}$$

The second term has a singularity at the origin. We can use Stokes's theorem on a disc centred at the origin to obtain

$$\vec{B}_2 = -2\pi N \delta(x^2 + y^2)\hat{z}. \tag{32}$$

We first consider the situation where we only have the magnetic field $\vec{B}_1$ and later reinstate the contribution from $\vec{B}_2$. Accordingly, we drop the $\vec{A}_2$ term from the Lagrangian and consider

$$\mathcal{L}_{\mathrm{Lan}} \equiv \mathcal{L}_{\mathrm{max}}^{(2)} - \dot{\vec{x}} \cdot \vec{A}_2 = \frac{1}{2}\frac{N}{L}\left(\dot{x}^2 + \dot{y}^2\right) + \frac{N}{L^2}(x\dot{y} - y\dot{x}). \tag{33}$$

The conjugate momenta are

$$p_x = \frac{N}{L}\left(\dot{x} - \frac{1}{L}y\right), \qquad p_y = \frac{N}{L}\left(\dot{y} + \frac{1}{L}x\right), \tag{34}$$

and the corresponding Hamiltonian is

$$\mathcal{H}_{\mathrm{Lan}} = \frac{L}{2N}\left(p_x^2 + p_y^2\right) + \frac{N}{2L}\frac{1}{L^2}(x^2 + y^2) - \frac{1}{L}\left(xp_y - yp_x\right). \tag{35}$$

Indeed, this is precisely the Hamiltonian for the Landau problem in the symmetric gauge of a particle of mass $\mu$ and (say positive) charge $q$ in a two-dimensional plane with a constant transverse magnetic field $B$, with the identifications

$$\mu = \frac{N}{L}, \qquad qB = \frac{2N}{L^2}. \tag{36}$$

---

Writing the three spaces as $\mathrm{AdS}_m \times S^n$ with $(m, n) = (4, 7), (5, 5), (7, 4)$, with $N$ units of $n$ form flux on the sphere $S^n$ of radius $L$ and expanding the brane action around the same point of the two-dimensional transverse geometry in $S^n$, (24), the Lagrangian takes the following form

$$\mathcal{L} = \frac{N}{L}\left(\frac{1}{2}\dot{\rho}^2 - L\dot{\varphi} + \frac{n-3}{2}L\rho^2\dot{\varphi} + \frac{1}{2}\rho^2\dot{\varphi}^2\right). \tag{26}$$

Thus, the only thing that changes depending on the dimension of the $(n-2)$-brane is the prefactor of the $\rho^2\dot{\varphi}$ term. We discuss the possible extension of our results for these two cases in a closing comment at the end of the article.

The solution to this quantum mechanical problem is well known, so we just outline its most relevant features and we refer the reader to [57–59] where it is introduced as a first step in the studies of the quantum Hall effect. Introducing the kinetic momentum operators

$$\pi_x = \mu \dot{x} = p_x + \frac{1}{2}qBy, \qquad \pi_y = \mu \dot{y} = p_y - \frac{1}{2}qBx, \tag{37}$$

the Landau Hamiltonian (35) takes the form

$$\mathcal{H}_{\text{Lan}} = \frac{1}{2\mu}(\pi_x^2 + \pi_y^2). \tag{38}$$

From the canonical commutation relations of the operators $x, y, p_x, p_y$ one finds

$$[\pi_x, \pi_y] = iqB, \tag{39}$$

and therefore the Hamiltonian (38) is algebraically the same as for the one-dimensional quantum harmonic oscillator. More explicitly, one can define the creation and annihilation operators

$$a = \frac{1}{\sqrt{2qB}}(\pi_x + i\pi_y), \qquad a^\dagger = \frac{1}{\sqrt{2qB}}(\pi_x - i\pi_y), \qquad [a, a^\dagger] = 1, \tag{40}$$

so that the Hamiltonian and the energy eigenvalues take the form

$$\mathcal{H}_{\text{Lan}} = \frac{qB}{\mu}\left(a^\dagger a + \frac{1}{2}\right), \qquad E_n = \frac{qB}{\mu}\left(n + \tfrac{1}{2}\right). \tag{41}$$

This effective dimensional reduction entails that the energy levels are degenerate. Each energy level, which contains infinitely many states, is called a Landau level. The groundstate level is called the lowest Landau level (LLL).

The degeneracy can be broken by exploiting the remaining symmetries of the system. That is, by labelling the states in each Landau level by their eigenvalue for some other quantum operator in the system which commutes with the Hamiltonian. The following operators

$$X = x + \frac{1}{qB}\pi_y, \qquad Y = y - \frac{1}{qB}\pi_x, \tag{42}$$

which are the quantum version of the position of the classical center of the orbit, satisfy the following commutation relation

$$[X, Y] = -\frac{i}{qB}, \tag{43}$$

and they commute with the Hamiltonian. We can define the following operator which measures the distance from the center of the classical orbit to the origin,

$$\mathcal{R}^2 = X^2 + Y^2 = \frac{2}{qB}\left(b^\dagger b + \frac{1}{2}\right), \tag{44}$$

where we have written it in terms of the following ladder operators,

$$b = \sqrt{\frac{qB}{2}}(X - iY), \qquad b^\dagger = \sqrt{\frac{qB}{2}}(X + iY), \qquad [b, b^\dagger] = 1. \tag{45}$$

The $\mathcal{R}^2$ operator can be used to label the different states in each Landau level. Defining $|0, 0\rangle$ as the state which is annihilated by $a$ and $b$, the states are labelled by two quantum numbers, $n, \ell \in \mathbb{N}_0$, $|n, \ell\rangle$, where $\ell$ is the eigenvalue for the $b^\dagger b$ operator. Later it will be used that the angular momentum operator,

$$\widehat{L} = xp_y - yp_x, \tag{46}$$

which is not gauge invariant, in the symmetric gauge takes the following form in terms of gauge invariant operators,

$$\widehat{L} = -\frac{1}{2qB}(\pi_x^2 + \pi_y^2) + \frac{1}{2}qB(X^2 + Y^2) = -a^\dagger a + b^\dagger b \,. \tag{47}$$

Therefore in the symmetric gauge we can label the states by their eigenvalues for the Hamiltonian and the angular momentum operator

$$\mathcal{H}_{\text{Lan}} |n, \ell\rangle = \frac{qB}{\mu}\left(n + \frac{1}{2}\right)|n, \ell\rangle \,, \qquad \widehat{L} |n, \ell\rangle = (-n + \ell)|n, \ell\rangle \,. \tag{48}$$

It is clear that the ground states are given by $|0, \ell\rangle$. The wavefunctions in the coordinate representation corresponding to the states in the lowest Landau level $|0, \ell\rangle$ are concisely written in complex coordinates,

$$\psi_{0,\ell}(z, \overline{z}) = C_\ell z^\ell e^{-|z|^2/4} \,, \tag{49}$$

where $C_\ell$ is a normalization constant and

$$z = \sqrt{qB}\left(x + iy\right), \qquad \overline{z} = \sqrt{qB}\left(x - iy\right). \tag{50}$$

So far we discussed the first part $A_1$ of the gauge field $A_1 + A_2$ in (30). The second part gives rise to the delta-functional magnetic field (32). In fact, this term can be absorbed into the above analysis by including a shift of the angular momentum (this is essentially the Aharonov-Bohm effect). We discuss this in Appendix B. The final result is that the spectrum of our problem is the same as that of the Landau problem with a shift of $N$ of the angular momentum.

## 3 Localization of the fluctuating giants

Our goal is to calculate the $\frac{1}{2}$-BPS index as a sum over all possible paths of $\frac{1}{2}$-BPS configurations of gravitons and D3-branes in $AdS_5 \times S^5$. As explained in the introduction we think of the branes as a gas of instantons in the Euclidean theory with periodic time and with all fields having periodic boundary conditions around the time circle. In the bulk theory we can write the index as the following functional integral,

$$I_N^{\text{bulk}}(q) = \sum_{m=0}^{\infty} \int_{\mathcal{M}(m)} d\mu_m \int d\phi_m \exp\bigl(S_{\text{brane+sugra}}(\phi_m; m, \mu_m)\bigr) q^R \,. \tag{51}$$

Here $m$ labels the number of branes and $\mu_m$ labels the moduli space $\mathcal{M}(m)$ of $m$ branes. Recall that the moduli space of one supersymmetric brane is the union of the space of giant gravitons (labelled by $\theta_0$ in (19) for one giant) and the space of dual giants. The fields $\phi_m$ denote the $\frac{1}{2}$-BPS fluctuations of the supergravity fields and of the collective coordinates of the branes in the ambient space. $S_{\text{brane+sugra}}$ is the combined action of the branes and the supergravity as a function of the moduli and of $\phi_m$. The charge $R$ is the $R$-charge used in the index (1), and acts on the space of $\frac{1}{2}$-BPS brane configurations and their fluctuations in string theory. The parameter $q$ is an external parameter with $|q| < 1$, which can be regarded as a background value for the gauge field in $AdS_5 \times S^5$ that couples to $R$.

The fluctuations of supergravity fields above can be thought of as a gas of gravitons surrounding and interacting with the branes. The fluctuations of the collective coordinates of the brane include a gauge field and scalars corresponding to motion in the transverse directions. The bosonic action for these fluctuations is given by the DBI+CS action (18). The brane preserves 16 and breaks 16 of the 32 supercharges of the background. Correspondingly, it

contains 16 fermions in its low energy fluctuation spectrum. The supersymmetric version of the action is constructed (for one brane) using a combination of superfield and $\kappa$-symmetry methods [60, 61].

As explained in the introduction, our strategy to calculate the path integral (51) is to localize it to $Q$-fixed points where $Q$ is one of the supercharges.[10] Equivalently, we construct a $Q$-exact action $QV$ and add it to the action with a parameter that we take to infinity. The sum over configurations in (51) thus reduces to a sum over the critical points of $QV$ of the classical brane action times the one-loop determinant of quadratic fluctuations. As we see below, the critical points are given by the maximal giants. The quadratic fluctuations factor into the quadratic fluctuations of the supergravity fields and those of the brane coordinates. The contribution of the former is simply that of a free gas of supergravitons. As we see below, the latter reduces to a supersymmetric version of the Landau problem.[11]

In Section 3.1 we briefly review the superalgebra in the presence of the branes. In Section 3.2 we describe the theory of small fluctuations of the branes including fermions, and set up the localizing action. In Sections 3.3 and 3.4, we calculate the critical points and one-loop determinants of this action.

## 3.1 Symmetry algebra

The superalgebra of the background $\text{AdS}_5 \times S^5$ is $psu(2,2|4)$. The bosonic sector of this algebra is $so(2,4) \oplus su(4)$ and there are 16 $Q$-supercharges and 16 $S$-supercharges.

Recall that the semiclassical $\frac{1}{2}$-BPS giant graviton wraps the $S^3 \subset S^5$, stays at the origin of $\text{AdS}_5$ and moves on a circle in $S^5$ parameterized by the angle $\phi$. Therefore it preserves $so(4) \oplus so(4) \oplus \mathbb{R}$ in the bosonic sector, where the two $so(4)$ algebras correspond to the spatial rotation of $\text{AdS}_5$ and the rotation of the $S^3 \subset S^5$, respectively. The generator of $\mathbb{R}$ is $H-R$, which act as $H = -i\partial_t$, $R = -i\frac{1}{L}\partial_\phi$ in the embedding coordinates. The corresponding generator in the Euclidean theory is $\partial_{t_E} - i\frac{1}{L}\partial_\phi$ and generates $u(1)$.

The algebra of the preserved symmetries is as follows. The $so(4)$ rotations of $S^3 \subset \text{AdS}_5$ is $su(2) \oplus su(2)$ with respective generators $J_\alpha^{\ \beta}$ and $\widetilde{J}_{\dot{\alpha}}^{\ \dot{\beta}}$. Similarly, the $so(4)$ rotations of $S^3 \subset S^5$ is $su(2) \oplus su(2)$ with respective generators $r_B^A$ and $\widetilde{r}_{\dot{A}}^{\dot{B}}$. Here, as usual, the indices $\alpha, \dot{\alpha}, A, \dot{A}$, $\beta, \dot{\beta}, B, \dot{B}$ label the doublet representation of $su(2)$. The supercharges $(Q_\alpha^A, S_B^{\ \beta})$, $(\widetilde{Q}_{\dot{A}\dot{\alpha}}, \widetilde{S}^{\dot{B}\dot{\beta}})$ fall into the representation $(\frac{1}{2}, 0, \frac{1}{2}, 0) \oplus (0, \frac{1}{2}, 0, \frac{1}{2})$ of the four $su(2)$ algebras. The embedding of the symmetries of the brane in the symmetries of $\text{AdS}_5 \times S^5$ is given in Appendix C. The non-zero anticommutators of the supercharges are

$$
\begin{aligned}
\left\{ Q_\alpha^A, S_B^{\ \beta} \right\} &= \delta_B^A J_\alpha^{\ \beta} - \delta_\alpha^{\ \beta} r_B^A + \frac{1}{2}\delta_\alpha^{\ \beta} \delta_B^A (H-R), \\
\left\{ \widetilde{Q}_{\dot{A}\dot{\alpha}}, \widetilde{S}^{\dot{B}\dot{\beta}} \right\} &= \delta_{\dot{A}}^{\dot{B}} \widetilde{J}_{\dot{\alpha}}^{\ \dot{\beta}} + \delta_{\dot{\alpha}}^{\ \dot{\beta}} \widetilde{r}_{\dot{A}}^{\dot{B}} + \frac{1}{2}\delta_{\dot{\alpha}}^{\ \dot{\beta}} \delta_{\dot{A}}^{\dot{B}} (H-R).
\end{aligned}
\tag{52}
$$

Compared to the notation in Appendix C, we have dropped the $'$ from the supercharges and set $R_z = R$. Note that each line of (52) is almost equivalent to $su(2|2)$ algebra with 6 bosonic and 8 fermionic generators. For a choice of its real form, this is observed to be the 2d chiral $\mathcal{N} = 4$ supersymmetry algebra (here $J$ generates the $sl(2)$ and $r$ generates the $su(2)$), which may potentially be useful. However, it is misleading to try to identify the brane theory

---

[10]A priori, the non-perturbative definition and convergence of the functional integral (51) is not completely clear. We thank the referee for emphasizing this point. As we see below, we can calculate the integral at weak gravitational coupling using localization with well-defined rules. The agreement of the answer with the boundary result, as expected from AdS/CFT, indicates that we are doing the correct thing.

[11]In general, the fluctuation analysis is non-trivial, as the supergravity fields are coupled to the fluctuations of the branes. We thank the referee for questioning this point. Localization allows us to analyze this in a weakly-coupled limit in which the gravitons and the open-string modes can be separately diagonalized and calculated.

as having a 2d $(4,4)$ algebra, especially because of the existence of the central extension $H{-}R$, which is the same in both lines.

In fact, this last term on the right-hand side of (52) is crucial. The theory of $\frac{1}{2}$-BPS states corresponds to the singlet excitations of the brane in the two $S^3$s, as can be seen from the algebra (52). Upon truncating to this sector we put the charges $J_\alpha^{\ \beta}$ and $r_B^A$ (and also $\widetilde{J}_{\dot\beta}^{\dot\alpha}$ and $\widetilde{r}_{\dot B}^{\dot A}$) to zero. Now we choose the supercharge $Q_-^1 \equiv \sqrt{2}Q$ and its conjugate $S_1^- \equiv \sqrt{2}\,\overline{Q}$, which, in the singlet sector, obey

$$\{Q,\overline{Q}\} = H{-}R = -i\left(\partial_t - \tfrac{1}{L}\partial_\phi\right) = \partial_{t_E} + i\tfrac{1}{L}\partial_\phi\,. \tag{53}$$

The idea we use below (which has been used before in other localization calculations) is that the fixed points of this subalgebra, in the Euclidean theory with appropriate reality conditions, are fixed points of $\partial_{t_E}$ and $i\partial_\phi$ separately. We achieve this by constructing an appropriate $Q$-exact action that we add to the action of the brane and then localize to the critical points of the deformation.

## 3.2 The theory of small fluctuations including fermions

In Section 2.3 we saw that the Lagrangian of quadratic fluctuations of the maximal giant is the sum of two Lagrangians: the first part governs the fluctuations of the brane in the $x_1, x_2$ (or, equivalently, the $\rho, \varphi$ directions), and the second part governs the fluctuations of the other bosonic directions ($r, S^3 \subset \text{AdS}_5$), and the gauge field on the brane. The latter part is gapped, while the former part is equivalent to a two-dimensional particle in a transverse magnetic field.

The low-energy theory on the brane contains 16 fermions corresponding to the broken supersymmetry generators. From the discussion in Section 3.1, we deduce that they transform in $(0,\frac{1}{2},0,\frac{1}{2}) \oplus (\frac{1}{2},0,\frac{1}{2},0)$ of $so(4) \oplus so(4)$. Now consider one of the supercharges $Q$ which obeys the algebra (53). This induces a pairing of the bosons $x_1, x_2$ and two of the fermions, which we call $\lambda_1, \lambda_2$, and between the other six bosonic fields and the remaining 6 fermions of one of the copies. Since the remaining 6 bosons are gapped, we expect the same for the corresponding 6 fermions due to $Q$-supersymmetry, and we expect a unique ground state of this gapped system. The ground states of the full system are a tensor product of this ground state of the gapped system with the $(x_1, x_2, \lambda_1, \lambda_2)$ system. In the doubled theory, again, 6 of the fermions will be gapped because of pairing by one of the $\widetilde{Q}$ supercharges.

The interesting subsystem is therefore described by $(x_1, x_2, \lambda_1, \lambda_2, \widetilde{\lambda}_1, \widetilde{\lambda}_2)$, which we now proceed to analyze. Here $\lambda_{1,2}(t)$, $\widetilde{\lambda}_{1,2}(t)$ are two real 1-dimensional Majorana fermions, i.e. Grassmann-valued fields. One should be able to derive the full supersymmetric Lagrangian from a direct analysis of the brane dynamics including fermions and analyzing its $\kappa$-symmetry, but we postpone such an analysis to the future. Instead we simply supersymmetrize the bosonic Lagrangian that we derived from the brane dynamics. This theory should be unique up to the quadratic order that we need.

First we focus on the $(x_1, x_2, \lambda_1, \lambda_2)$ system without doubling. The supersymmetrization of (33) is given by the following simple Lagrangian,[12] see e.g. [63–66]:

$$\frac{L}{N}\mathcal{L}_{\max}^{\text{susy}(2)} = \sum_{i=1}^{2}\left(\frac{1}{2}\dot{x}_i^2 - \frac{1}{2}i\dot{\lambda}_i\lambda_i + \dot{x}_i A_i\right) - i\,B\,\lambda_1\lambda_2\,, \tag{54}$$

with

$$A_1 = -\frac{1}{L}x_2\,, \qquad A_2 = \frac{1}{L}x_1\,, \qquad B = \frac{\partial A_2}{\partial x_1} - \frac{\partial A_1}{\partial x_2} = \frac{2}{L}\,. \tag{55}$$

---

[12]One can think of this Lagrangian as a dimensional reduction of a supersymmetric Chern-Simons theory in three dimensions [62]. The bilinear coupling of the fermions to $B$ is also familiar from this description.

This Lagrangian is invariant under $\mathcal{N} = 1$ supersymmetry transformations [63]. In fact it is easy to check that the Lagrangian (54) is invariant under the following $\mathcal{N} = 2$ supersymmetry transformations

$$
\begin{aligned}
\delta_1 x_1 &= \lambda_1, & \delta_1 \lambda_1 &= -i\,\dot{x}_1, & \delta_1 x_2 &= \lambda_2, & \delta_1 \lambda_2 &= -i\,\dot{x}_2, \\
\delta_2 x_1 &= \lambda_2, & \delta_2 \lambda_2 &= -i\,\dot{x}_1, & \delta_2 x_2 &= -\lambda_1, & \delta_2 \lambda_1 &= i\,\dot{x}_2.
\end{aligned}
\tag{56}
$$

The above variations obey the algebra (calling the new time coordinate $T$ for now)

$$
(\delta_1)^2 = -2\,i\,\partial_T, \qquad (\delta_2)^2 = -2\,i\,\partial_T, \qquad \{\delta_1, \delta_2\} = 0.
\tag{57}
$$

Now that we have constructed the supersymmetric theory of maximal brane fluctuations, we need to map the charges of the original theory onto this new theory. From the change of coordinates (24), we see that there is a corresponding shift of the time-translations as

$$
(t, \phi) = (T, \frac{T}{L} - \varphi) \quad \Longrightarrow \quad \partial_\varphi = -\partial_\phi, \qquad -i\partial_T = -i\left(\partial_t - \frac{1}{L}\partial_\varphi\right).
\tag{58}
$$

Now comparing the right-hand side of $(\delta_1)^2$ in (57) with the relation $\{Q, \overline{Q}\} = H - R$ in the original SYM theory, we see that we should identify $R = -\frac{i}{L}\partial_\varphi$ on the space of fluctuations of the maximal giant. Note that the reality conditions on the field fluctuations of this space is the one relevant for the fluctuations of the saddle in the Euclidean theory, and is different from the physical brane fluctuations.

In fact, the Lagrangian (54) can be written as a $Q$-exact term. Choosing $Q = \delta_1$ given in (56) we have

$$
\begin{aligned}
V_1 &= \frac{1}{2}(\lambda_1 Q \lambda_1 + \lambda_2 Q \lambda_2) = -i\frac{1}{2}(\dot{x}_1 \lambda_1 + \dot{x}_2 \lambda_2), \\
V_2 &= -i\frac{1}{L}(x_2 Q x_1 - x_1 Q x_2) = -i\frac{1}{L}(x_2 \lambda_1 - x_1 \lambda_2).
\end{aligned}
\tag{59}
$$

It is easy to check that

$$
\begin{aligned}
Q V_1 &= \frac{1}{2}\left(\dot{x}_1^2 + \dot{x}_2^2 - i\dot{\lambda}_1 \lambda_1 - i\dot{\lambda}_2 \lambda_2\right), \\
Q V_2 &= -\frac{1}{L}\left(\dot{x}_2 x_1 - x_2 \dot{x}_1 - 2i\lambda_1 \lambda_2\right),
\end{aligned}
\tag{60}
$$

so that, with $V = V_1 + V_2$,

$$
Q V = \frac{L}{N}\mathcal{L}_{\max}^{\mathrm{susy}(2)}.
\tag{61}
$$

Thus, in order to obtain the fixed points of $Q$, we can study the critical points of the bosonic part of this Lagrangian, which is precisely the Landau Lagrangian (33).

Before doing so, we include the doubling of fermions in the fluctuations of the maximal giant. Now we have the fermionic fields $\lambda_j, \widetilde{\lambda}_j, j = 1, 2$, and the action is given by

$$
\frac{L}{N}\mathcal{L}_{\max}^{\mathrm{susy}\text{-}2} = \sum_{j=1}^{2}\left(\frac{1}{2}\dot{x}_j^2 + \dot{x}_j A_j\right) - \frac{i}{2}\sum_{j=1}^{2}\dot{\lambda}_j \lambda_j - iB\lambda_1\lambda_2 - \frac{i}{2}\sum_{j=1}^{2}\dot{\widetilde{\lambda}}_j \widetilde{\lambda}_j - iB\widetilde{\lambda}_1\widetilde{\lambda}_2,
\tag{62}
$$

with $A_1, A_2, B$ given, as before, in (55). We already saw that the fields $(x_j, \lambda_j)$ are paired so that the first line of (62) is supersymmetric under the action of the supercharge $Q$. The fermions $\widetilde{\lambda}_j$, on the other hand, are inert under the action of $Q$. The total system, therefore, is supersymmetric, but in a sense that is slightly unusual in that one set of fermions do not have bosonic superpartners.[13]

---

[13]The full theory of the brane discussed in Section 3.1 has a symmetry that exchanges the $Q$ and $\widetilde{Q}$ supercharges and, simultaneously, the $\lambda$ and $\widetilde{\lambda}$ fermions. Accordingly, (62) is also invariant under a choice of $\widetilde{Q}$, in which case the fields $(x_j, \widetilde{\lambda}_j)$ are paired and $\lambda_j$ are singlets.

### 3.3 Critical points

Now we study the critical points of the supersymmetric Lagrangian $QV$ discussed in the above subsection. Upon setting the fermions to zero, we obtain the bosonic Lagrangian (33). We then perform the Euclidean continuation $t = -it_E$ and impose that the field configurations respect periodicity in $t_E$ with period $\beta$. The functional integral in the Euclidean theory is weighted by $\exp\left(\int dt_E \mathcal{L}^{\mathrm{E}(2)}_{\mathrm{max}}\right)$,[14] with (here $\dot{} = \frac{d}{dt_E}$)

$$-\frac{L}{N}\mathcal{L}^{\mathrm{E}(2)}_{\mathrm{max}} = \frac{1}{2}\dot{\rho}^2 + \frac{1}{2}\rho^2\dot{\varphi}^2 + \frac{1}{2}(iB)\rho^2\dot{\varphi}. \tag{63}$$

Since the Lagrangian is independent of $\varphi$, the angular momentum is a conserved quantity, i.e.,

$$\frac{L}{N}J_{\mathrm{E}} = \frac{L}{N}\frac{\partial \mathcal{L}^{\mathrm{E}(2)}_{\mathrm{max}}}{\partial \dot{\varphi}} = -\rho^2\dot{\varphi} - \frac{1}{2}iB\rho^2, \qquad \frac{d}{dt_E}J_{\mathrm{E}} = 0. \tag{64}$$

We now use the conserved quantity $J_E$ in order to eliminate $\dot{\varphi}$ and reduce the problem to a one-dimensional problem for $\rho(t_E)$. The resulting equation is

$$\rho^3\ddot{\rho} = \frac{\rho^4}{L^2} + \frac{L^2}{N^2}J_{\mathrm{E}}^2. \tag{65}$$

Since $\rho$ is a radial variable in the physical problem, we continue to demand that $\rho(t_E)$ is a non-negative real field in the Euclidean problem. The equation of motion (65) then implies that $J_E$ is real and that $\ddot{\rho} \geq 0$. The periodicity of $\rho(t_E)$ implies that the only consistent option is $\ddot{\rho} = 0$, and therefore $\rho = 0$ and $J_E = 0$.

We see from the discussion in Appendix B that the theory of the brane fluctuations is equivalent to a 2d particle in a constant transverse magnetic field with angular momentum shifted by $N$ units. Thus we see that the above critical point corresponds to the proper angular momentum being equal to $N$, which is the maximal giant graviton solution. In other words, the functional integral localizes to maximal giant gravitons. We note that the set of critical points does not include the dual giants.[15]

### 3.4 One-loop fluctuation determinant

In this subsection we calculate the determinant of quadratic fluctuations of the maximal giant. As in any instanton calculation, the fluctuation determinant is calculated over the fluctuations of background fields (supergravitons) and of the collective coordinates or open string fluctuations of the branes. The localization analysis allows us to go to the weakly-gravitational coupled limit, in which the supergravitons and the collective coordinates can be separately diagonalized. The first problem of integrating over the fields of supergravity is equivalent to calculating the index (1) on the Hilbert space of the gas of supergravitons, and is given by $I_{\mathrm{sugra}}(q) = I_\infty(q)$.

Our formula for the bulk index thus reduces to the following localized form,

$$I_N^{\mathrm{bulk}}(q) = I_{\mathrm{sugra}}(q)\sum_{m=0}^{\infty}\int d\phi_m \exp\left(S^{(2)}_{\mathrm{brane}}(\phi_m; m)\right)q^R, \tag{66}$$

where $\phi_m$ denotes all the fluctuations of the maximal giants with periodic boundary conditions on the Euclidean time circle, with quadratic action $S^{(2)}_{\mathrm{brane}}$. The functional integral (66) is equivalent to the original index (1) evaluated on the Hilbert space of small fluctuations of the

---

[14]We take the convention that the functional integral in the Lorentzian theory is weighted by $\exp(iS)$.

[15]This is in contrast with the duality between giants and dual giants in the Lorentzian theory [2, 46, 67].

branes. In the following discussion we analyze one brane ($m = 1$), and in the next section we move to multiple branes.

The small fluctuations split into massive and massless modes. The massless modes for $m = 1$ are governed by the Lagrangian (62). All the other fluctuations of the brane are massive and $R$ commutes with the Hamiltonian governing them. Therefore, by the usual pairing argument, they do not contribute to the index. Thus we are left with calculating the index (1) in the theory of $(x_1, x_2, \lambda_1, \lambda_2, \tilde{\lambda}_1, \tilde{\lambda}_2)$ governed by the Lagrangian (62). This is a well-defined quantum system without gravity. Therefore, we can calculate the quadratic fluctuation determinant as a functional integral or, equivalently, as a Hamiltonian index on the Hilbert space of fluctuations. Here we use the Hamiltonian formalism as it is easier, and also physically instructive. It would also be interesting to calculate this determinant in the functional integral formalism. Our approach is to canonically quantize this system and calculate the index by explicitly listing the ground states of this system.

As in Section 3.2, we start by considering the theory of $(x_1, x_2, \lambda_1, \lambda_2)$ with the following Lagrangian,

$$\frac{L}{N} \mathcal{L}_{\text{max}}^{\text{susy}(2)} = \frac{1}{2}\dot{x}_1^2 + \frac{1}{2}\dot{x}_2^2 + \frac{1}{2}B(-\dot{x}_1 x_2 + \dot{x}_2 x_1) - \frac{1}{2}i\dot{\lambda}_1\lambda_1 - \frac{1}{2}i\dot{\lambda}_2\lambda_2 - iB\lambda_1\lambda_2. \tag{67}$$

The corresponding Hamiltonian is

$$H_{\text{SLan}} = \frac{1}{2}\left(p_1 + \frac{1}{2}Bx_2\right)^2 + \frac{1}{2}\left(p_2 - \frac{1}{2}Bx_1\right)^2 + \frac{i}{2}B[\lambda_1, \lambda_2]. \tag{68}$$

The canonical commutation relations are

$$[x_i, p_j] = i\delta_{ij}, \qquad \{\lambda_i, \lambda_j\} = \delta_{ij}. \tag{69}$$

The algebra of the bosonic sector is unchanged, so the bosonic sector can be solved exactly as for the original Landau problem. In the fermionic sector we see that the anticommutation relations for $\lambda_i$ (69) define a two-dimensional Clifford algebra. We use the following representation,

$$\lambda_1 = \frac{1}{\sqrt{2}}\sigma_1, \qquad \lambda_2 = \frac{1}{\sqrt{2}}\sigma_2, \qquad [\lambda_1, \lambda_2] = i\sigma_3, \tag{70}$$

where $\sigma_i, i = 1, 2, 3$ are the Pauli matrices. The Hamiltonian in this representation reads

$$H_{\text{SLan}} = \frac{1}{2}\left(p_1 + \frac{1}{2}Bx_2\right)^2 + \frac{1}{2}\left(p_2 - \frac{1}{2}Bx_1\right)^2 - \frac{1}{2}B\sigma_3, \tag{71}$$

which is a bosonic and a fermionic oscillator, both of frequency $B$.

The Hamiltonian now takes the following form

$$H_{\text{SLan}} = B\left(a^\dagger a + \frac{1}{2} - \frac{1}{2}\sigma_3\right). \tag{72}$$

Labelling the two eigenstates of $\sigma_3$ as

$$\frac{1}{2}\sigma_3 |s\rangle = s |s\rangle, \qquad s = \pm\frac{1}{2}, \tag{73}$$

the energy spectrum of the Hamiltonian is given by

$$E_{n,s} = B\left(n + \frac{1}{2} - s\right). \tag{74}$$

The ground states have $n = 0$, $s = \frac{1}{2}$ and have zero energy.

The angular momentum operator, including the fermionic spin, is given by

$$\widehat{L} = x_1 p_2 - x_2 p_1 - \frac{1}{2} i [\lambda_1, \lambda_2]. \tag{75}$$

Using the Pauli matrices representation of the $so(2)$ Clifford algebra we have

$$\widehat{L} = x_1 p_2 - x_2 p_1 + \frac{1}{2} \sigma_3. \tag{76}$$

The bosonic part of the angular momentum operator for the Landau problem was discussed in Section 2.3. The states can be completely described by three quantum numbers $|n, \ell, s\rangle$ and have the following eigenvalues of the Hamiltonian and angular momentum operators, with $n, \ell = 0, 1, 2, \dots$ and $s = \pm \frac{1}{2}$,

$$H |n, \ell, s\rangle = B \left( n + \frac{1}{2} - s \right) |n, \ell, s\rangle,$$
$$\widehat{L} |n, \ell, s\rangle = \left( -n + \ell + s \right) |n, \ell, s\rangle. \tag{77}$$

Now we consider the following Lagrangian with all four fermions,

$$\frac{L}{N} \mathcal{L}_{\max}^{\text{susy-2}} = \frac{1}{2} \dot{x}_1^2 + \frac{1}{2} \dot{x}_2^2 + \frac{1}{2} B (-\dot{x}_1 x_2 + \dot{x}_2 x_1) - \frac{1}{2} i \dot{\lambda}_1 \lambda_1 - \frac{1}{2} i \dot{\lambda}_2 \lambda_2 - i B \lambda_1 \lambda_2$$
$$- \frac{1}{2} i \dot{\widetilde{\lambda}}_1 \widetilde{\lambda}_1 - \frac{1}{2} i \dot{\widetilde{\lambda}}_2 \widetilde{\lambda}_2 - i B \widetilde{\lambda}_1 \widetilde{\lambda}_2. \tag{78}$$

The Hamiltonian takes the form

$$H_{\text{SLan-2}} = \frac{1}{2} \left( p_1 + \frac{1}{2} B x_2 \right)^2 + \frac{1}{2} \left( p_2 - \frac{1}{2} B x_1 \right)^2 + \frac{i}{2} B [\lambda_1, \lambda_2] + \frac{i}{2} B [\widetilde{\lambda}_1, \widetilde{\lambda}_2]. \tag{79}$$

Then, the non-zero canonical commutation relations for the position, momenta, and fermionic operators are

$$[x_i, p_j] = i \delta_{ij}, \qquad \{\lambda_i, \lambda_j\} = \delta_{ij}, \qquad \{\widetilde{\lambda}_i, \widetilde{\lambda}_j\} = \delta_{ij}. \tag{80}$$

The algebra of the bosonic sector is unchanged, so the bosonic sector can be solved exactly as for the original Landau problem. For the fermionic sector, as before, we use the representation in terms of two sets of Pauli matrices:

$$\lambda_i = \frac{1}{\sqrt{2}} \sigma_i, \qquad \{\sigma_i, \sigma_j\} = 2 \delta_{ij}, \qquad \widetilde{\lambda}_i = \frac{1}{\sqrt{2}} \widetilde{\sigma}_i, \qquad \{\widetilde{\sigma}_i, \widetilde{\sigma}_j\} = 2 \delta_{ij}, \tag{81}$$

so that we now have a product of two two-dimensional Clifford algebras. The Hamiltonian is

$$H_{\text{SLan-2}} = B \left( a^\dagger a + \frac{1}{2} - \frac{1}{2} (\sigma_3 + \widetilde{\sigma}_3) \right), \tag{82}$$

where

$$i \sigma_3 = [\lambda_1, \lambda_2], \qquad i \widetilde{\sigma}_3 = [\widetilde{\lambda}_1, \widetilde{\lambda}_2]. \tag{83}$$

The angular momentum operator is

$$\widehat{L} = x_1 p_2 - x_2 p_1 + \frac{1}{2} (\sigma_3 + \widetilde{\sigma}_3), \tag{84}$$

and the spectrum takes the form

$$H |n, \ell, s, \widetilde{s}\rangle = B \left( n + \frac{1}{2} - (s + \widetilde{s}) \right) |n, \ell, s, \widetilde{s}\rangle,$$
$$\widehat{L} |n, \ell, s, \widetilde{s}\rangle = \left( -n + \ell + s + \widetilde{s} \right) |n, \ell, s, \widetilde{s}\rangle, \tag{85}$$

where $s, \widetilde{s} = \pm\frac{1}{2}$ are the eigenvalue of the operator $\frac{1}{2}\sigma_3, \frac{1}{2}\widetilde{\sigma}_3$, i.e.,

$$\frac{1}{2}\sigma_3 |s,\widetilde{s}\rangle = s |s,\widetilde{s}\rangle\,, \qquad \frac{1}{2}\widetilde{\sigma}_3 |s,\widetilde{s}\rangle = \widetilde{s} |s,\widetilde{s}\rangle\,. \tag{86}$$

Again, notice that the states $\left|n,\ell,-\frac{1}{2},\widetilde{s}\right\rangle$ and $\left|n+1,\ell,\frac{1}{2},\widetilde{s}\right\rangle$ have the same eigenvalues for $H$ and $\widehat{L}$, and the same is true for $\left|n,\ell,s,-\frac{1}{2}\right\rangle$ and $\left|n+1,\ell,s,\frac{1}{2}\right\rangle$. Since as we later show, the states in each pair have opposite fermion number, they cancel and do not contribute to the index, and the only states that contribute are the ones in the LLL with both $s, \widetilde{s}$ eigenvalues positive.

The LLL are the least energy states,

$$H\left|0,\ell,\frac{1}{2},\frac{1}{2}\right\rangle = -\frac{B}{2}\left|0,\ell,\frac{1}{2},\frac{1}{2}\right\rangle\,, \tag{87}$$

which have the following eigenvalues for the angular momentum operator

$$\widehat{L}\left|0,\ell,\frac{1}{2},\frac{1}{2}\right\rangle = (\ell+1)\left|0,\ell,\frac{1}{2},\frac{1}{2}\right\rangle\,, \tag{88}$$

with $\ell \geq 0$.

Now we can assemble our results. We had identified the $R$ charge operator as $R = -i\partial/\partial\varphi$ on the space of fluctuations, where it takes the form

$$R = -i\frac{1}{L}\partial_\varphi = \widehat{L}\,. \tag{89}$$

On the ground states of the LLL, this takes the values $R = 1, 2, \ldots$, each with degeneracy 1. In other words,

$$\mathrm{Tr}_{\mathrm{LLL}}\, q^R = \sum_{n=1}^{\infty} q^n = \frac{q}{1-q}\,, \qquad \text{for } |q| < 1\,. \tag{90}$$

**The sign of the index**

Since we are computing a Witten index, we should track the eigenvalues of the states under the fermion number operator $(-1)^F$. According to the AdS/CFT duality, this operator is simply the fermion number in the bulk. In the theory of quadratic fluctuations that is relevant to us here, the fermion number is a tensor product of the fermion number operator acting on the supergravity fields and on the brane fluctuations. We have already analyzed the former in calculating the index of supergravity fields. The brane fermion number at the quadratic level naturally descends from the current

$$F = \frac{1}{2}\sum_{\text{fermions }\psi}\int d^4\sigma\, [\overline{\psi(\sigma)}, \psi(\sigma)]\,, \tag{91}$$

which is built up of the free fermionic fluctuations in its worldvolume theory. For our quantum mechanical system this is given by

$$F = -\frac{i}{2}([\lambda_1, \lambda_2] + [\widetilde{\lambda}_1, \widetilde{\lambda}_2]) = \frac{1}{2}(\sigma_3 + \widetilde{\sigma}_3)\,. \tag{92}$$

The fermionic sector is spanned by the states $|s,\widetilde{s}\rangle$ with $s, \widetilde{s} = \pm\frac{1}{2}$, with $(-1)^F = (-1)^{s+\widetilde{s}}$.

As stated earlier, the states $\left|n,\ell,-\frac{1}{2},\widetilde{s}\right\rangle$ and $\left|n+1,\ell,\frac{1}{2},\widetilde{s}\right\rangle$ have the same eigenvalues for $H$ and $\widehat{L}$, and moreover they have opposite $(-1)^F$ eigenvalue, and therefore their contribution to the index cancels. The same applies for the states of the form $\left|n,\ell,s,-\frac{1}{2}\right\rangle$ and $\left|n+1,\ell,s,\frac{1}{2}\right\rangle$. The only case where there is no cancellation is for the LLL, with *all* states fermionic.[16] This concludes the computation of the Witten index for the quadratic fluctuations of one maximal giant, the complete answer being

$$\text{Tr}_{\mathcal{H}^{(2)}_{\text{max. giant}}}(-1)^F q^R = -\frac{q}{1-q}. \tag{93}$$

We note that we have used a simple, natural version of fermion number in the quadratic sector of the brane system,[17] which should be universal, in order to define the bulk Witten index. This gives the answer expected from AdS/CFT.

## 4 Multiple giants

In this section we address the problem of multiple giants. We first rewrite the bosonic Landau Hamiltonian

$$\mathcal{H}_{\text{Lan}} = \frac{1}{2}\left(p_x^2 + p_y^2\right) + \frac{1}{2}L^2\left(x^2 + y^2\right) - \frac{1}{L}\left(xp_y - yp_x\right), \tag{94}$$

in complex coordinates

$$z = \frac{1}{\sqrt{2}}(x + iy), \qquad \overline{z} = \frac{1}{\sqrt{2}}(x - iy), \tag{95}$$

to obtain

$$\mathcal{H}_{\text{Lan}} = p_z p_{\overline{z}} + \frac{1}{L^2}z\overline{z} - \frac{i}{L}\left(zp_z - \overline{z}p_{\overline{z}}\right), \tag{96}$$

where $p_z$ and $p_{\overline{z}}$ are the conjugate momenta to $z$ and $\overline{z}$. The associated Lagrangian is

$$\mathcal{L}_{\text{Lan}} = \dot{z}\dot{\overline{z}} - \frac{i}{L}\left(\dot{z}\overline{z} - \dot{\overline{z}}z\right), \tag{97}$$

and the canonical momenta

$$p_z = \dot{\overline{z}} - \frac{i}{L}\overline{z}, \qquad p_{\overline{z}} = \dot{z} + \frac{i}{L}z. \tag{98}$$

The generalization to $m$ coincident giants consists in promoting the scalar fields describing the transverse fluctuations of the branes to $m \times m$ matrix-valued fields transforming in the adjoint of $U(m)$. The fluctuations of one giant with two bosonic degrees of freedom is described now by two matrix-valued scalar fields. Two different hermitian matrices, in general, cannot be simultaneously diagonalized. However, the Landau problem is special since the Hamiltonian for two bosonic degrees of freedom can be reduced to the Hamiltonian of a one-dimensional harmonic oscillator. Moreover, in complex variables the annihilation operators take the form such that the functions which are annihilated by them are any holomorphic function times a Gaussian. This holomorphicity is what makes the groundstate spectrum for the matrix problem solvable. (See e.g. [48], [14] for more details and related discussion.)

---

[16]The same conclusion can be reached by noting that there is an additional $\widetilde{Q}$ susy in the system, cf Footnote [13] and repeating the arguments for the theory of two fermions.

[17]However, as noted earlier, the rigorous derivation of the full fermionic sector remains to be done. We thank the referee for emphasizing this point.

Essentially, the problem reduces to $m$ decoupled eigenvalues, with energies $1, 2, \ldots m$, each of the type that we discussed in the previous subsection. Therefore the answer is given by the product of (93) with $q \mapsto q^j$, $j = 1, 2, \ldots m$. In the rest of the section we write out some of the details of the reduction to the holomorphic sector.

Consider the non-abelian generalization of (97) which describes the bosonic part of $m$ coincident maximal giants,

$$\mathcal{L}_m = \text{Tr}\left( \dot{Z}\dot{Z}^\dagger - \frac{i}{L}\left( \dot{Z}Z^\dagger - \dot{Z}^\dagger Z \right) \right). \tag{99}$$

where $Z$ and its adjoint $Z^\dagger$ are $m \times m$ complex matrices. Writing out the indices we get

$$
\begin{aligned}
\mathcal{L}_m &= \text{Tr}\left( \sum_{j=1}^m \left( \dot{Z}_{ij}(\dot{Z}^\dagger)_{jk} - \frac{i}{L}\left( \dot{Z}_{ij}(Z^\dagger)_{jk} - (\dot{Z}^\dagger)_{ij}Z_{jk} \right) \right) \right) \\
&= \sum_{i,j=1}^m \left( \dot{Z}_{ij}(\dot{Z}^\dagger)_{ji} - \frac{i}{L}\left( \dot{Z}_{ij}(Z^\dagger)_{ji} - (\dot{Z}^\dagger)_{ij}Z_{ji} \right) \right) \\
&= \sum_{i,j=1}^m \left( \dot{Z}_{ij}\dot{\overline{Z}}_{ij} - \frac{i}{L}\left( \dot{Z}_{ij}\overline{Z}_{ij} - \dot{\overline{Z}}_{ji}Z_{ji} \right) \right) \\
&= \sum_{i,j=1}^m \left( \dot{Z}_{ij}\dot{\overline{Z}}_{ij} - \frac{i}{L}\left( \dot{Z}_{ij}\overline{Z}_{ij} - \dot{\overline{Z}}_{ij}Z_{ij} \right) \right).
\end{aligned}
\tag{100}
$$

That is, it consists of $m^2$ copies of the Lagrangian of the Landau problem with $2m^2$ degrees of freedom. The canonical momenta take the form

$$P_{ij} = \frac{\partial \mathcal{L}_m}{\partial \dot{Z}_{ij}} = \dot{\overline{Z}}_{ij} - \frac{i}{L}\overline{Z}_{ij}, \qquad \overline{P}_{ij} = \frac{\partial \mathcal{L}_m}{\partial \dot{\overline{Z}}_{ij}} = \dot{Z}_{ij} + \frac{i}{L}Z_{ij}, \tag{101}$$

and the Hamiltonian reads

$$\mathcal{H}_m = \sum_{i,j=1}^m \left( P_{ij}\overline{P}_{ij} + \frac{1}{L^2}Z_{ij}\overline{Z}_{ij} - \frac{i}{L}\left( Z_{ij}P_{ij} - \overline{Z}_{ij}\overline{P}_{ij} \right) \right). \tag{102}$$

From the canonical commutation relations

$$[Z_{ij}, P_{ij}] = i, \qquad [\overline{Z}_{ij}, \overline{P}_{ij}] = i, \qquad \text{since} \quad P_{ij} = -i\frac{\partial}{\partial Z_{ij}}, \quad \overline{P}_{ij} = -i\frac{\partial}{\partial \overline{Z}_{ij}}, \tag{103}$$

we can write the Hamiltonian in the following form

$$\mathcal{H}_m = \sum_{i,j=1}^m \left( \dot{\overline{Z}}_{ij}\dot{Z}_{ij} + \frac{1}{L} \right) = \text{Tr}\, \dot{Z}^\dagger \dot{Z} + \frac{m^2}{L}, \tag{104}$$

with the operators appearing in the Hamiltonian satisfying the following commutation relations

$$[\dot{Z}_{ij}, \dot{\overline{Z}}_{ij}] = \frac{2}{L}. \tag{105}$$

Therefore, the Hamiltonian (104) describes $m^2$ one-dimensional harmonic oscillators, or the matrix harmonic oscillator described by an $m \times m$ Hermitian matrix, which we can write as

$$\mathcal{H}_m = \frac{2}{L}\left( -\frac{1}{2}\Delta + \text{Tr}(W^2) \right), \tag{106}$$

for $W$ an $m \times m$ hermitian matrix and $\Delta$ the kinetic operator for the matrix problem

$$\Delta = \sum_{i=1}^{m} \frac{\partial^2}{\partial W_{ii}^2} + \frac{1}{2} \sum_{1 \leq i < j \leq m} \frac{\partial^2}{\partial \operatorname{Re} W_{ij}^2} + \frac{\partial^2}{\partial \operatorname{Im} W_{ij}^2}. \tag{107}$$

We can now use the classical results of [68], where it was shown that enforcing the $U(m)$ invariance of the system, it gets reduced to the dynamics of its $m$ eigenvalues which behave as fermionic fields. More concretely, the wavefunction becomes a function of the eigenvalues $w_i$ of $W$ only, being an eigenfunction of the sum of $m$ harmonic oscillators

$$\frac{2}{L} \left( -\frac{1}{2} \sum_{i=1}^{m} \frac{\partial^2}{\partial w_i^2} + w_i^2 \right), \tag{108}$$

with an added prefactor

$$\prod_{1 \leq i < j \leq m} (w_i - w_j), \tag{109}$$

which makes it antisymmetric under the exchange of any two eigenvalues. To write the solution in terms of our original variables $Z, Z^\dagger$, we write the following representation for the annihilation operators

$$\dot{Z}_{ij} = \overline{P}_{ij} - \frac{i}{L} Z_{ij} = -i \left( \frac{\partial}{\partial \overline{Z}_{ij}} + \frac{1}{L} Z_{ij} \right) = -i e^{-\frac{z_{ij} \overline{z}_{ij}}{L}} \frac{\partial}{\partial \overline{Z}_{ij}} e^{\frac{z_{ij} \overline{z}_{ij}}{L}}. \tag{110}$$

The groundstate wavefunctions annihilated by these operators therefore take the form

$$\widetilde{\Psi}_{LLL}(Z, Z^\dagger) = C F(Z) e^{-\frac{\operatorname{Tr}\left(zz^\dagger\right)}{L}}, \tag{111}$$

where $C$ is a normalization constant and $F(Z)$ is a holomorphic function of $Z$. The angular momentum operator in the symmetric gauge in the $Z, \overline{Z}$ coordinates is

$$J = \frac{i}{L} \sum_{i,j=1}^{m} \left( Z_{ij} P_{ij} - \overline{Z}_{ij} \overline{P}_{ij} \right) = \frac{1}{L} \sum_{i,j=1}^{m} \left( Z_{ij} \frac{\partial}{\partial Z_{ij}} - \overline{Z}_{ij} \frac{\partial}{\partial \overline{Z}_{ij}} \right). \tag{112}$$

The Gaussian function $e^{-\frac{\operatorname{Tr}\left(zz^\dagger\right)}{L}}$ is annihilated by it and therefore the angular momentum operator on the LLL wavefunctions gets reduced to its holomorphic part

$$J|_{\text{LLL}} = \frac{1}{L} \sum_{i,j=1}^{m} Z_{ij} \frac{\partial}{\partial Z_{ij}}. \tag{113}$$

Once we have restricted ourselves to the holomorphic part of the problem we can diagonalize the angular momentum operator to

$$J|_{\text{LLL}} = \frac{1}{L} \sum_{i=1}^{m} z_i \frac{\partial}{\partial z_i}, \tag{114}$$

where $z_i$ are the eigenvalues of $Z$, and the wavefunction for the groundstate of the Hamiltonian (104) takes the form

$$\Psi_{LLL}(z_i, Z^\dagger) = c f(z) \prod_{1 \leq i < j \leq m} (z_i - z_j) e^{-\frac{\operatorname{Tr}\left(zz^\dagger\right)}{L}}, \tag{115}$$

where $c$ is a normalization constant, $\prod_{1 \leq i < j \leq m}(z_i - z_j)$ is the Vandermonde determinant appearing from the diagonalization and $f(z)$ is a symmetric function of the eigenvalues $z_i$.

Having found the general form for the groundstate wavefunction, the only thing left is to compute the spectrum of groundstates labelled by their $J$-eigenvalue. As explained in Section 2.3, for the case of a single particle in the LLL the states can be labelled by their angular momentum eigenvalue in the symmetric gauge, with the spectrum being the same as for the harmonic oscillator. For $m$ coincident giants, the problem is similar to the one of studying $m$ fermionic particles in the LLL, and one can see that the wavefunction (115) is similar to the Laughlin wavefunction. In the latter case, the groundstate consists of $m$ fermionic oscillators, and therefore the $J$-charge spectrum should look like the one of $m$ fermionic oscillators with frequencies being $1, \ldots, m$ multiples of the single-giant frequency.

This is indeed the case. First, the following derivative of the Vandermonde determinant just gives the number of terms in the product

$$\left(\sum_{i=1}^{m} z_i \partial_i\right) \prod_{1 \leq i < j \leq m}(z_i - z_j) = \frac{m(m-1)}{2} \prod_{1 \leq i < j \leq m}(z_i - z_j). \tag{116}$$

Second, the set of wavefunctions of the form (115) which are eigenvectors of (114) is spanned by the symmetric function $f(z)$ being a monomial symmetric polynomial of the $m$ eigenvalues $z_i$. The number of monomial symmetric polynomials in $m$ variables of total degree $\ell$ is given by the number of partitions of $\ell$ into at most $m$ parts. If we include the shift by 1 induced by supersymmetrization of the Hamiltonian so that the angular momentum of each particle starts at 1, it will be given by all partitions of $\ell$ into exactly $m$ parts, $p_m(\ell)$, which has generating function

$$\sum_{\ell=1}^{\infty} p_m(\ell) q^\ell = q^m \prod_{n=1}^{m} \frac{1}{1 - q^n}. \tag{117}$$

Then the number of states contributing to the index from the $m \times m$ matrix sector is

$$q^{m(m-1)/2} q^m \prod_{n=1}^{m} \frac{1}{1 - q^n}. \tag{118}$$

The contribution has to be graded by the number $(-1)^F$. The groundstates for the single giant case are all fermionic, having all $(-1)^F = -1$. Since $(-1)^F$ is multiplicative, for the case of $m$ coincident giants the operator $(-1)^F$ is given by $m$ times the single giant one, $(-1)^m$, and therefore the graded contribution to the index of quadratic fluctuations is

$$(-1)^m \frac{q^{m(m+1)/2}}{(q)_m}. \tag{119}$$

Finally, we still have to add the $R$-charge of the $m$ maximal giants, with each one shifting the value by $N$ units so that the final contribution for $m$ coincident maximal giants to the index is

$$(-1)^m \frac{q^{m(m+1)/2}}{(q)_m} q^{mN}. \tag{120}$$

### A comment on the M2 and M5-brane theories

As described in Footnote 9, the bosonic Lagrangian for the quadratic fluctuations of a single maximal giant graviton in $\text{AdS}_4 \times S^7$, which is an M5-brane wrapping $S^5 \subset S^7$, and of a single maximal giant graviton in $\text{AdS}_7 \times S^4$, which is an M2-brane wrapping $S^2 \subset S^4$, are both essentially the same as the fluctuations of the maximal D3-giant graviton in $\text{AdS}_5 \times S^5$. Therefore,

it appears that for a single maximal giant graviton one can extend the analysis of Section 3 to these two cases to produce the same result.[18] This is actually consistent with the $\frac{1}{2}$-BPS indices of the corresponding boundary theories of $N$ M2-branes or M5-branes. Indeed, the $\frac{1}{2}$-BPS indices for $k = 1$ ABJM theory with gauge group $U(N) \times U(N)$ and for the $A_{N-1}$ 6d $(2, 0)$ theory coincide with the $\frac{1}{2}$-BPS index for $\mathcal{N} = 4$ SYM with $U(N)$ gauge group given in (2), as shown in e.g. [69] and [70], respectively. This predicts that the full giant-graviton expansion agrees for all these cases.

The analysis of this section for multiple branes will also go through if the matrix problem reduces to decoupled eigenvalues with charges $1, 2, 3, \ldots$. The non-abelian theories for M2- as well as M5-branes involve a matrix-valued holomorphic scalar field as in the D3-brane theory. For the M2 case, the matrix-valued scalar field is in a bifundamental representation of the gauge group, and these transformations can be used to diagonalize the matrix. The complete non-abelian structure of the M5-branes is not known, but in any string theory limit, as in [70], the adjoint action of the gauge group can be used to diagonalize the matrix. It would be interesting to work out the details of the non-abelian structure and verify the prediction of the giant-graviton expansion for these theories.

## Acknowledgments

It is a pleasure to thank Chi-Ming Chang, Sumit Das, Nadav Drukker, Juan Maldacena, Gautam Mandal, Shiraz Minwalla, George Papadopoulos, Nemani Suryanarayana, and Edward Witten for useful and enjoyable discussions. We would like to thank the Isaac Newton Institute for Mathematical Sciences, Cambridge, for support and hospitality during the programme Black holes: bridges between number theory and holographic quantum information where work on this paper was undertaken. We would like to thank the Asia Pacific Center for Theoretical Physics (APCTP), Korea, for support and hospitality during the programme Quantum Black Holes, Quantum Information and Quantum Strings where work on this paper was undertaken.

**Funding information** S.M. acknowledges the support of the J. Robert Oppenheimer Visiting Professorship at the Institute for Advanced Study, Princeton, USA and the STFC grants ST/T000759/1, ST/X000753/1. This work was supported by EPSRC grant no. EP/R014604/1. G.E. is supported by the STFC grant ST/V506771/1 and by an educational grant offered by the A. G. Leventis Foundation.

## A  Derivation of the $\frac{1}{2}$-BPS expansion

It is useful to introduce the $q$-Pochhammer symbols

$$(x; q)_n = \prod_{j=1}^{n} (1 - x \, q^{j-1}), \qquad (q)_n = (q; q)_n, \tag{A.1}$$

in terms of which we can write $I_N(q) = 1/(q)_N$. We have

$$\frac{I_N(q)}{I_\infty(q)} = \frac{(q; q)_\infty}{(q; q)_N} = (q^{N+1}; q)_\infty. \tag{A.2}$$

The identity (3) follows by substituting $x = q^{N+1}$ in the identity

$$(x; q)_\infty = \sum_{m=0}^{\infty} a_m x^m, \qquad a_m = (-1)^m \frac{q^{\binom{m}{2}}}{(q)_m}. \tag{A.3}$$

---

[18]We would like to thank the referee for raising this point.

The formula for the coefficients $a_m$ can be easily proved by induction in $m$ [71], by noting that $a_0 = 1$ and by using the functional equation

$$(x;q)_\infty = (1-x)(qx;q)_\infty \, ,\tag{A.4}$$

both of which follow from the definition (A.1) of the $q$-Pochhammer symbol.

## B  Flux tube and spectral flow

In this section we study the effect of reintroducing the gauge potential term $\vec{A}_2$, which we dropped in Section 2.3, on the spectrum obtained for the Landau problem. We explain how the term $\vec{A}_2$, which describes an infinitely thin solenoid at the center of the plane carrying $N$ units of flux, leaves the spectrum of the Hamiltonian invariant, but produces a change in the spectrum of the $J$ operator at each Landau level. We do so in two ways, by looking at how the Hamiltonian operators for the two systems are related, and by seeing the change as an extra Aharonov-Bohm phase in the wavefunctions due to the added flux.

As we saw in Section 2.3, the gauge potential

$$\vec{A} = \frac{N}{L^2}(x\hat{y} - y\hat{x})\left(1 - \frac{L^2}{x^2 + y^2}\right),\tag{B.1}$$

naturally splits into two parts as

$$\vec{A} = \vec{A}_1 + \vec{A}_2, \qquad \vec{A}_1 = \frac{N}{L^2}(x\hat{y} - y\hat{x}), \qquad \vec{A}_2 = -N\frac{x\hat{y} - y\hat{x}}{x^2 + y^2}.\tag{B.2}$$

Upon discarding $\vec{A}_2$, the Landau Hamiltonian (35) becomes, in polar coordinates,

$$H_0 = \frac{1}{2N/L}\left[-\frac{1}{\rho}\frac{\partial}{\partial\rho}\left(\rho\frac{\partial}{\partial\rho}\right) + \left(-\frac{i}{\rho}\frac{\partial}{\partial\varphi} - \frac{N}{L^2}\rho\right)^2\right].\tag{B.3}$$

As we discussed around Equation (32), the gauge potential $\vec{A}_2$ is flat everywhere outside the origin, and corresponds to an infinitely thin solenoid, i.e. a magnetic field of strength $2\pi N$ along a vertical line through the origin pointing downwards. Reinstating this term, we obtain the full Hamiltonian to be

$$H = \frac{1}{2N/L}\left[-\frac{1}{\rho}\frac{\partial}{\partial\rho}\left(\rho\frac{\partial}{\partial\rho}\right) + \left(-\frac{i}{\rho}\frac{\partial}{\partial\varphi} - \frac{N}{L^2}\rho + \frac{N}{\rho}\right)^2\right].\tag{B.4}$$

Thus we see that the ground states of the full Hamiltonian are given by the ground states of the Landau problem, but with the angular momentum shifted by $N$ units. This is simply the angular momentum of the ground state of the giant.

The change in eigenvalue for the $J$ operator when we reintroduce the $\vec{A}_2$ term can be seen as a consequence of the Aharonov-Bohm effect. Consider the Landau Lagrangian (33). As a particle goes around a loop in the presence of a gauge field $\vec{A}$, its wavefunction will acquire the following Aharonov-Bohm phase

$$\phi_{AB} = \oint \vec{A} \cdot d\vec{s} = 2\pi\Phi \, ,\tag{B.5}$$

where $\Phi$ is the total magnetic flux enclosed by the loop.

Consider the state in the lowest Landau level $|0,\ell\rangle$, with wavefunction (49) given in the units and variables of Section 2.3,

$$\psi_{0,\ell} = C_\ell z^\ell e^{-|z|^2/4} = C_\ell \rho^\ell e^{i\ell\varphi} e^{-\rho^2/4}.$$ (B.6)

Now imagine turning on a small flux $\Delta\Phi$ through an infinitely thin solenoid placed at the origin. By performing a (singular) gauge transformation we can eliminate the flux so that the wavefunction changes as $\psi \mapsto e^{-i\Delta\Phi\varphi}\psi$. Therefore $\psi$ is single-valued only when $\Delta\Phi$ is an integer. Upon changing the flux $\Delta\Phi$ from 0 to $-N$ adiabatically and then performing a gauge transformation to eliminate it, the eigenstate in the lowest Landau level with eigenvalue $\ell$ for the angular momentum operator $J$ transforms to an eigenstate with eigenvalue $\ell - N$. In other words, the change in angular momentum is seen to be a consequence of the $-N$ units of flux through an infinitely thin solenoid. This phenomenon is similar to Laughlin's pumping argument in the studies of the quantum Hall effect (see, e.g. [58,59]). In our case it just comes out of the quadratic fluctuations of a maximal giant. It can be put on firmer mathematical grounds by defining an index which computes the difference between the projection to the LLL operators before and after applying the singular gauge transformation [72].

## C  Superalgebra of a giant graviton in AdS$_5 \times S^5$

In this appendix we review the superalgebra that is preserved by the $\frac{1}{2}$-BPS giant graviton. This symmetry algebra is also studied in [21].

The algebra of the background AdS$_5 \times S^5$ is $psu(2,2|4)$. The bosonic sector of this algebra is $so(2,4) \oplus su(4)$ and there are 16 $Q$-supercharges and 16 $S$-supercharges. The generators of the algebra are

- $J_\alpha^\beta$, with $\alpha,\beta = +,-$ and $\widetilde{J}_{\dot\beta}^{\dot\alpha}$, with $\dot\alpha,\dot\beta = +,-$, which are the generators of Lorentz transformations,

- $P_{\alpha\dot\alpha}$, with $\alpha = +,-$ and $\dot\alpha = \dot+,\dot-$, which is the generator of translations,

- $H$, the generator of dilations,

- $K^{\dot\alpha\alpha}$, with $\alpha = +,-$ and $\dot\alpha = \dot+,\dot-$, which is the generator of special conformal transformations,

- $R_J^I$, with $I,J = 1,\ldots,4$, which is the generator of $R$-symmetry, noting that it has 15 independent components since it is traceless,

- $Q_\alpha^I, S_I^\alpha$, with $I = 1,\ldots,4$ and $\alpha = +,-$, and $\widetilde{Q}_{I\dot\alpha}, \widetilde{S}^{I\dot\alpha}$, with $I = 1,\ldots,4$ and $\dot\alpha = \dot+,\dot-$, which are the supercharges.

The Cartan charges $(E, j, \widetilde{j}, R_1, R_2, R_3)$ are given by

$$\begin{aligned}
E &= H, & j &= J_+^+ = -J_-^-, & \widetilde{j} &= \widetilde{J}_{\dot+}^{\dot+} = -\widetilde{J}_{\dot-}^{\dot-}, \\
R_1 &= R_1^1 - R_2^2, & R_2 &= R_2^2 - R_3^3, & R_3 &= R_3^3 - R_4^4.
\end{aligned}$$ (C.1)

The Cartan charges of $su(4)_R$ $(R_1, R_2, R_3)$ and $so(6)_R$ $(R_x, R_y, R_z)$, which are isomorphic, are related by

$$\begin{aligned}
R_1 &= R_y + R_z, \\
R_2 &= R_x - R_y, \\
R_3 &= R_y - R_z.
\end{aligned}$$ (C.2)

Therefore we also have the following relations

$$R_x = R^1_1 + R^2_2, \qquad R_y = R^1_1 + R^3_3, \qquad R_z = R^1_1 + R^4_4. \tag{C.3}$$

The semiclassical $\frac{1}{2}$-BPS giant graviton wraps the $S^3 \subset S^5$, stays at the origin of AdS$_5$ and moves on a circle in $S^5$. Introducing the brane breaks translation and special conformal symmetry, so the theory with the brane preserves $so(4) \oplus so(4) \oplus \mathbb{R}$ in the bosonic sector, where the two $so(4)$ algebras correspond to the spatial rotation of AdS$_5$ and the rotation of the $S^3 \subset S^5$, respectively. The wrapped D3-brane breaks half and preserves half of the supersymmetries. The preserved supercharges are 8 $Q$-supercharges and 8 $S$-supercharges satisfying $H = R_z$. These are $Q^1_\pm, Q^4_\pm, \widetilde{Q}_{2\pm}, \widetilde{Q}_{3\pm}$ and their conjugates $S^\pm_1, S^\pm_4, \widetilde{S}^{2\pm}, \widetilde{S}^{3\pm}$.

It is useful to give a notation consistent with the $su(2) \oplus su(2)$ algebra:

$$Q'^1_\pm = Q^1_\pm, \qquad Q'^2_\pm = Q^4_\pm, \qquad \widetilde{Q}'_{1\pm} = \widetilde{Q}_{2\pm}, \qquad \widetilde{Q}'_{\dot{2}\pm} = \widetilde{Q}_{3\pm},$$
$$S'^\pm_1 = S^\pm_1, \qquad S'^\pm_2 = S^\pm_4, \qquad \widetilde{S}'^{1\pm} = \widetilde{S}^{2\pm}, \qquad \widetilde{S}'^{\dot{2}\pm} = \widetilde{S}^{3\pm}, \tag{C.4}$$

$$r^1_1 = \frac{1}{2}(R^1_1 - R^4_4), \qquad r^1_2 = R^4_4, \qquad r^2_1 = R^1_1, \qquad r^2_2 = -\frac{1}{2}(R^1_1 - R^4_4),$$
$$\widetilde{r}^{\dot{1}}_{\dot{1}} = \frac{1}{2}(R^2_2 - R^3_3), \qquad \widetilde{r}^{\dot{1}}_{\dot{2}} = R^3_3, \qquad \widetilde{r}^{\dot{2}}_{\dot{1}} = R^2_2, \qquad \widetilde{r}^{\dot{2}}_{\dot{2}} = -\frac{1}{2}(R^2_2 - R^3_3). \tag{C.5}$$

The full preserved algebra is obtained by starting with the algebra of the background theory and setting to zero the generators of the broken symmetries, i.e. $P$, $K$, and the broken 8 $Q$-supercharges and the 8 $S$-supercharges. In the following (anti)commutation relations, the Greek letters take values $+, -$ and the Latin letters take values $1, 2$. The preserved supercharges satisfy the following anticommutation relations

$$\left\{ Q'^I_\alpha, S'^\beta_J \right\} = \delta^I_J J^\beta_\alpha - \delta^\beta_\alpha r^I_J + \frac{1}{2} \delta^\beta_\alpha \delta^I_J (H - R_z),$$
$$\left\{ \widetilde{Q}'_{I\dot{\alpha}}, \widetilde{S}'^{J\dot{\beta}} \right\} = \delta^j_i \widetilde{J}^{\dot{\beta}}_{\dot{\alpha}} + \delta^{\dot{\beta}}_{\dot{\alpha}} \widetilde{r}^j_i + \frac{1}{2} \delta^{\dot{\beta}}_{\dot{\alpha}} \delta^j_i (H - R_z). \tag{C.6}$$

The $so(4) \cong su(2) \oplus su(2)$ subalgebra corresponding to the spatial rotations of AdS$_5$ is given by

$$[J^\beta_\alpha, J^\delta_\gamma] = \delta^\beta_\gamma J^\delta_\alpha - \delta^\delta_\alpha J^\beta_\gamma, \qquad [\widetilde{J}^{\dot{\alpha}}_{\dot{\beta}}, \widetilde{J}^{\dot{\gamma}}_{\dot{\delta}}] = \delta^{\dot{\alpha}}_{\dot{\delta}} \widetilde{J}^{\dot{\gamma}}_{\dot{\beta}} - \delta^{\dot{\gamma}}_{\dot{\beta}} \widetilde{J}^{\dot{\alpha}}_{\dot{\delta}}. \tag{C.7}$$

The supercharges transform as a doublet under this subalgebra, i.e.,

$$\left[ J^\beta_\alpha, Q'^I_\gamma \right] = \delta^\beta_\gamma Q'^I_\alpha - \frac{1}{2} \delta^\beta_\alpha Q'^I_\gamma, \qquad \left[ \widetilde{J}^{\dot{\alpha}}_{\dot{\beta}}, \widetilde{Q}'_{I\dot{\gamma}} \right] = \delta^{\dot{\alpha}}_{\dot{\gamma}} \widetilde{Q}'_{I\dot{\beta}} - \frac{1}{2} \delta^{\dot{\alpha}}_{\dot{\beta}} \widetilde{Q}'_{I\dot{\gamma}},$$
$$\left[ J^\beta_\alpha, S'^\gamma_I \right] = -\delta^\gamma_\alpha S'^\beta_I + \frac{1}{2} \delta^\beta_\alpha S'^\gamma_I, \qquad \left[ \widetilde{J}^{\dot{\alpha}}_{\dot{\beta}}, \widetilde{S}'^{I\dot{\gamma}} \right] = -\delta^{\dot{\gamma}}_{\dot{\beta}} \widetilde{S}'^{I\dot{\alpha}} + \frac{1}{2} \delta^{\dot{\alpha}}_{\dot{\beta}} \widetilde{S}'^{I\dot{\gamma}}. \tag{C.8}$$

The $so(4) \cong su(2) \oplus su(2)$ subalgebra corresponding to the rotation of the $S^3 \subset S^5$ is given by

$$\left[ r^I_J, r^K_L \right] = \delta^K_J r^I_L - \delta^I_L r^K_J, \qquad \left[ \widetilde{r}^i_j, \widetilde{r}^{\dot{k}}_{\dot{l}} \right] = \delta^{\dot{k}}_{\dot{j}} \widetilde{r}^i_{\dot{l}} - \delta^i_{\dot{l}} \widetilde{r}^{\dot{k}}_j, \tag{C.9}$$

where $r^I_J$ and $\widetilde{r}^i_j$ are traceless. The supercharges transform as a doublet under this subalgebra, with the following commutation relations

$$\left[ r^I_J, Q'^K_\alpha \right] = \delta^K_J Q'^I_\alpha - \frac{1}{2} \delta^I_J Q'^K_\alpha, \qquad \left[ \widetilde{r}^i_j, \widetilde{Q}'_{\dot{K}\dot{\alpha}} \right] = -\delta^i_{\dot{K}} \widetilde{Q}'_{\dot{J}\dot{\alpha}} + \frac{1}{2} \delta^i_j \widetilde{Q}'_{\dot{K}\dot{\alpha}},$$
$$\left[ r^I_J, S'^\alpha_K \right] = -\delta^I_K S'^\alpha_J + \frac{1}{2} \delta^I_J S'^\alpha_K, \qquad \left[ \widetilde{r}^i_j, \widetilde{S}'^{\dot{K}\dot{\alpha}} \right] = \delta^{\dot{k}}_j \widetilde{S}'^{i\dot{\alpha}} - \frac{1}{2} \delta^i_j \widetilde{S}'^{\dot{K}\dot{\alpha}}. \tag{C.10}$$

The Cartan charges $(\ell, j, \widetilde{j}, r_1, \widetilde{r}_1)$ are given by

$$
\begin{aligned}
\ell &= H - R_z = E - R_z, & j &= J_+^+ = -J_-^-, & \widetilde{j} &= \widetilde{J}_+^{\dot{+}} = -\widetilde{J}_-^{\dot{-}}, \\
r_1 &= r_1^1 - r_2^2 = R_x + R_y, & \widetilde{r}_1 &= \widetilde{r}_1^{\dot{1}} - \widetilde{r}_2^{\dot{2}} = R_x - R_y.
\end{aligned}
\tag{C.11}
$$

We observe that the algebra splits into two identical parts, with one being generated by $(J, Q', S', r, H - R_z)$ and the other being generated by $(\widetilde{J}, \widetilde{Q}', \widetilde{S}', \widetilde{r}, H - R_z)$. So the bosonic part of the algebra is two copies of $su(2) \oplus su(2) \oplus \mathbb{R}$ with 4 $Q$-supercharges and 4 $S$-supercharges each.

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
