# Peer review of "The giant graviton expansion in $AdS_5 \times S^5$"

_SciPost Physics, doi:SciPost Phys. 17, 098 (2024)_

## Round 2 · Referee Report · Anonymous (Referee 1) · 2024-4-17

Report
The authors study the index of the half-BPS giant gravitons. While it has been already given in the literature, they propose an alternative derivation from the bulk string theory. They claim that as the effective Lagrangian of the giant gravitons obtained from the D3-brane action can be viewed as the 1d supersymmetric quantum mechanical Lagrangian for the Landau problem describing a particles in a two-dimensional plane with a constant magnetic flux, the index can be evaluated as its Witten index which has contributions from the lowest Landau level. Though the result of the index is already known, their approach would be potentially interesting and useful for experts working on the holographic duality in the context of string theory and supersymmetric gauge theory. However, there are several parts with which I am confused in the draft and I would like the authors to carefully improve.
Recommendation
Ask for major revision
Sameer Murthy on 2024-04-23 [id 4442]
We thank the referee for all the useful comments. Our responses are written below, and we hope this removes the confusions that are mentioned in the report. -- The authors.
Main comments 1. In Bullet 1 the referee states that the various quantities in the functional integral (3.1) are unclear. 2. In Bullets 2 and 1, the referee asks how the calculation of the above functional integral is done in practice, and questions the use of the Hamiltonian formalism. 3. In Bullets 3 and 4, the referee questions our treatment of fermions and fermion number.
-- The space of $\frac12$-BPS configurations in the bulk string theory on AdS$_5 \times S^5$ has been studied in the references quoted in the paper. When the bulk gravitational coupling is weak, these configurations are identified with $\frac12$-BPS graviton fluctuations around empty AdS5, fluctuations of $\frac12$-BPS D3-branes in AdS$_5 \times S^5$, and multiple combinations of them.
-- The functional integral (3.1) consists of a sum over an arbitrary number $m=0,1,2,...$ of $\frac12$-BPS D3-branes, with an integral over their moduli spaces, and with a further integral over $\frac12$-BPS field fluctuations around any given point in moduli space. For $m$ branes one has a combination of $m$ such objects with the usual identifications of identical particles in the quantum theory.
-- The charge $R$ is the $R$-charge used in the index (1.1), and acts on the above space of $\frac12$-BPS brane configurations and their fluctuations in the bulk string theory. This is the usual identification of R-charge between the bulk and the boundary in AdS/CFT, which identifies it with a certain rotation of the $S^5$. The parameter $q$ is an external parameter with $|q|<1$, which can be regarded as a background value for the gauge field in AdS$_5 \times S^5$ that couples to $R$.
-- We agree that, a priori, the non-perturbative definition and convergence of the functional integral (3.1) in the string theory is not completely clear. The point of the paper is that we can calculate the integral at weak gravitational coupling using localization, where the rules are well-defined. The agreement of the answer with the boundary result, as expected from AdS/CFT, indicates that we are doing the correct thing.
2a. Calculation of the bulk functional integral -- For one brane, the moduli space is the possible configurations of $\frac12$-BPS giants or dual giants. The space of giants is labelled by the size of $S^3 \subset S^5$, or equivalently, the angle on $S^2$. The space of dual giants is labelled by the size of $S^3 \subset AdS_5$. This has been well-studied in the references, e.g. Ref.[2]. The Euclidean functional integral localizes onto the space of maximal giants.
-- Following the statement of localization, we now have to calculate the action and determinant of quadratic fluctuations around the space of $m$ maximal giants and sum them up. The fluctuations include the fluctuations of the gravitational fields, as well as the collective coordinates of the brane or, equivalently, the open-string fields. In general, the fluctuation analysis is non-trivial in general where the supergravity fields are coupled to the fluctuations of the branes. However, the nice thing about localization is that it allows us to analyze this in a weakly-coupled limit in which the gravitons and the collective coordinates can be separately diagonalized.
2b. Hamiltonian vs functional integral -- Finally, we need to calculation of the determinant of the quadratic operator. For the reason explained above, this factorizes into the determinant of the supergravity fields and the determinant of the collective coordinates. Each of these two determinants is a separately well-defined quantum problem (without gravity!). Therefore, we can calculate these determinants as a functional integral or as a Hamiltonian problem on a well-defined Hilbert space of fluctuations. We have chosen to perform the Hamiltonian version since it is easier.
-- The first problem of integrating over the fields of supergravity is equivalent to quantizing a gas of gravitons. The corresponding index is the multi-graviton index. This is well-known and we simply quote the result in the paper. The second problem is the supersymmetric Landau problem which we discuss in detail in the paper.
-- We agree that it is an interesting problem to perform the same determinants (and reproduce our answer consistent with AdS/CFT!) by functional integral methods---either by explicitly calculating the eigenmodes, or perhaps by (again) using localization. This would be a nice follow-up to our paper, but not a logical necessity to complete the calculation.
3a. Treatment of fermionic action The referee's point here is a good one. We agree that a fully first-principles derivation of the action should begin by analyzing the $\kappa$-symmetry fixed D3-brane action in the curved space (or perhaps, depending on one's point of view, from string field theory). Here we do something more modest. We analyze the bosonic problem in curved space, analyze it in the Lagrangian and the Hamiltonian formalisms, and then supersymmetrize it minimally according to the symmetries of the problem. We believe that, at the level of quadratic fluctuations that we need in the paper, the supersymmetric terms should be universal. This is a fairly standard approach to constructing supersymmetric actions from bottom-up. In our opinion, the $\kappa$-symmetry analysis that the referee wishes to do is at the level of a separate paper (which, of course, would be interesting!).
3b. Treatment of fermion number -- By the AdS/CFT correspondence, the fermion number in the bulk follows from the fermion number in the boundary. Once we have constructed a brane theory, the fermion number in the free limit is naturally defined. In our context, it should simply be the fermion number of the theory of fluctuations of the branes. Since we have an essentially free theory at quadratic order, with extended supersymmetry, we actually have a fermion number current from which follows the $\mathbb{Z}_2$ symmetry.
-- One nice point about the whole above discussion is that the identifications that we make following the standard rules of AdS/CFT and the known space of branes leads to the correct giant graviton formula---with a very simple identification of the negative signs, i.e. as the bulk fermion number. In particular, we do not need any auxiliary construct to explain it.

---

## Round 3 · Referee Report · Anonymous (Referee 1) · 2024-6-9

Report

The authors improved the manuscript. However, there is the unclear point which I am still confused with.

If the method which authors propose is correct, one should be able to derive the indices of the half-BPS M-brane giants. As the authors discuss in footnote 9, the Lagrangian of the half-BPS M2-brane giant and that of the half-BPS M5-brane giant take the same form as that of the half-BPS D3-brane giant and the only different thing is the prefactor of the $\rho^2\dot{\varphi}$. So it is mentioned that the same procedure can be performed in a unified way. Then their method will lead to the essentially same indices of the half-BPS M-brane giants. However, this does not seem to be consistent since the indices of the half-BPS M-brane giants will be significantly different from the index of the half-BPS D3-brane giant. As the half-BPS D3-brane giant index was already known in the literature, the potential significance of this work will be to provide a new pathway to giant graviton indices in other setup. Hence I would like the authors to explain how their method can lead to the correct indices of the half-BPS M-brane giants in a transparent manner. If their method does not work for the M-brane giants, their method or interpretation may contain a fatal flaw.

Recommendation

Ask for major revision

  • validity: -
  • significance: -
  • originality: -
  • clarity: -
  • formatting: -
  • grammar: -

Author:  Sameer Murthy  on 2024-07-15  [id 4619]

(in reply to Report 1 on 2024-06-09)

This is a very useful comment for which we thank the referee. Indeed, Footnote 9 in the paper says that the small fluctuations of the half-BPS branes in the maximally supersymmetric cases (AdS5 x S5, AdS4 x S7, and AdS7 x S4) are all essentially the same, up to the coefficient of the term corresponding to the B-field that the fluctuations couple to. The referee asks about the interpretation in the dual field theory.

In fact, the result agrees with the half-BPS indices for the M2 and M5 brane theories found in the literature. The half-BPS index is shown to coincide with the one for N=4 SYM for ABJM theory in https://arxiv.org/pdf/0904.4605 (see Section 2.2 and, in particular, Equation (2.25) and the text surrounding it) and for the M5-brane theory in https://arxiv.org/pdf/1210.0853 (see Equation (4.4)).

We are happy to include this in the paper with a bit more detail (appropriately citing the referee for the question).

---

## Round 3 · Referee Report · Anonymous (Referee 2) · 2024-6-17

Report

This paper computes the expansion of the index of the 1/2 BPS operators in $\mathcal{N}=4$ SYM , known as the Giant Graviton expansion, from the bulk point of view.

The main object is the Euclidean functional integral over the configuration space of 1/2 BPS giants and dual giants. The integral localizes to a sum over fluctuations around arbitrary number of maximal giant gravitons. An interesting observation is that the excitations of one maximal giant are governed by the supersymmetric version of the Landau problem. The authors compute the index of these excitations which recovers the first term in the expansion $Z_N/Z_\infty$ and extend this to multiple giants.

The paper is an interesting contribution since it illustrates how each term (including the signs) in the expansion of the 1/2 BPS index directly arises from giant gravitons in the bulk and opens potential to computing the expansion on the bulk side in other examples. Therefore, I am happy to recommend this paper for publication in SciPost.

There are a few small typos: - footnote 5: construction "of the" Hilbert space - above (2.36): "take" the form - appendix C title: "a" giant graviton

Recommendation

Publish (easily meets expectations and criteria for this Journal; among top 50%)

  • validity: -
  • significance: -
  • originality: -
  • clarity: -
  • formatting: -
  • grammar: -

Author:  Sameer Murthy  on 2024-07-31  [id 4669]

(in reply to Report 2 on 2024-06-17)

We thank the referee for the report. We will make the small corrections that have been listed.
-- The authors.

---

## Round 3 · Author Response

The reply to the referee (posted as a comment below the report) contains the responses to the major comments of the referee, the corresponding changes in the text are indicated in the List of changes.
The responses to the minor comments are contained in the List of changes.

---

## Round 3 · List of Changes

Major comments: Our reply to the referee contains the responses to the referee report. Correspondingly, the introductory part of Section 3 has been expanded and Footnotes 10 and 11 have been added to clarify and emphasize some of the points.

Minor comments: 1. « On page 9, “in these coordinate" should be “in these coordinates” ». Response: We did not find “in these coordinate” on Page 9. The phrase “in these coordinates” does appear just above Eqn 2.23, and it seems to be correct.

  1. « In several occasions, e.g. eq.(2.41), (3.25), L is introduced as a momentum operator, but it may cause a confusion with the radius of AdS5 for which it was firstly used. » Response: We have changed the angular momentum operator to \widehat{L}.

  2. « While μ is used for mass of particle in eq.(2.31), it again appears as moduli in eq.(3.1) as the moduli of m branes. If they are not related with each other, it would be better to use a different notation. » Response: We have changed the notation of moduli to μ_m (since they depend on m anyway). We have also cleaned up the text below Equation 3.1.

  3. « On page 13, it is mentioned “this is the 2d chiral N = 4 supersymmetry algebra”. Why is the supersymmetry algebra associated with 2d spacetime? The effective Lagrangian is 1d quantum mechanics (rather than 2d field theory) whose fields only depend on time coordinate. On the other hand, if it is N = (4, 4) discussed below, it will be non-chiral supersymmetry rather than chiral supersymmetry. » Response: This was simply an observation. Indeed, as we had already remarked just below that sentence: “However, it is misleading to try to identify the brane theory as having a 2d (4, 4) algebra, especially because of the existence of the central extension H − R, which is the same in both lines.” We have modified the text accordingly by explicitly inserting the word “observation" in order to avoid a confusion along the above lines.

---

## Round 4 · Referee Report · Anonymous (Referee 1) · 2024-8-31

Report

The authors improved the manuscript by responding to the comments. I am happy to recommend the publication to SciPost.

Recommendation

Publish (meets expectations and criteria for this Journal)

---

## Round 4 · Author Response

Changes made as requested.

---

## Round 4 · List of Changes

Added comment on the M2 and M5-brane theories at the end of the paper. Typos corrected as listed on the SciPost page.

---

## Editorial Decision

published